

**Air quality and health benefits from ultra-low emission**
**control policy indicated by continuous emission monitoring:**
**A case study in the Yangtze River Delta region, China**
Yan Zhang[1], Yu Zhao[1, 2*], Meng Gao[3], Xin Bo[4], Chris P. Nielsen[5]
1. State Key Laboratory of Pollution Control and Resource Reuse and School of the
Environment, Nanjing University, 163 Xian in Ave., Nanjing, Jiangsu 210023, China.
2. Jiangsu Collaborative Innovation Center of Atmospheric Environment and
Equipment Technology (CICAEET), Nanjing University of Information Science and
Technology, Jiangsu 210044, China.
3. Department of Geography, State Key Laboratory of Environmental and Biological
Analysis, Hong Kong Baptist University, Hong Kong SAR, China.
4. The Appraisal Center for Environment and Engineering, Ministry of Environmental
Protection, Beijing 100012, China.
5. Harvard-China Project on Energy, Economy and Environment, John A. Paulson
School of Engineering and Applied Sciences, Harvard University, 29 Oxford St,
Cambridge, MA 02138, USA.
*Corresponding author: Yu Zhao
Phone: 86-25-89680650; email: *yuzhao@nju.edu.cn*





## Abstract

To evaluate improved emission estimation from online monitoring data, we applied the Models-3/CMAQ (Community Multi-scale Air Quality) system to simulate the air quality of the Yangtze River Delta (YRD) region using two emission inventories without/with incorporated data from continuous emission monitoring systems (CEMS) at coal-fired power plants (Cases 1 and 2), respectively. The normalized mean biases (NMBs) of annual $SO_2$, $NO_2$, $O_3$ and $PM_{2.5}$ concentrations between observations and simulations in Case 2 were -3.1%, 56.3%, -19.5% and -1.4%, all smaller in absolute value than those in Case 1, at 8.2%, 68.9%, -24.6% and 7.6%, respectively. The results indicate that incorporation of CEMS data in the emission inventory helped reduce the biases between simulation and observation and can better reflect the actual sources of regional air pollution. Based on the CEMS data, the air quality changes and corresponding health impacts were quantified for different implementation levels of China's recent "ultra-low" emission policy. If only the coal-fired power sector met the requirement, the simulated differences in the monthly $SO_2$, $NO_2$, $O_3$ and $PM_{2.5}$ concentrations compared to those of Case 2, our base case for policy comparisons, were less than 7% for all pollutants. The result implies only a minor benefit of ultra-low emission control if implemented in the power sector alone, attributed to its limited contribution to total emissions in the YRD after years of pollution control in the sector (11%, 7% and 2% of $SO_2$, $NO_X$ and primary particle matter (PM), respectively). If the ultra-low emission policy was enacted at both power plant and industrial boilers, the simulated $SO_2$, $NO_2$ and $PM_{2.5}$ concentrations compared to the base case were 33%-64%, 16%-23% and 6%-22% lower respectively, depending on the month (January, April, July and October 2015). Combining CMAQ and the Integrated Exposure Response (IER) model, we further estimated that 305 deaths and 874 years of life loss (YLL) attributable to $PM_{2.5}$ exposure could be avoided with the implementation of the ultra-low emission policy in the power sector in the YRD region. The analogous values would be much higher, at 10,651 deaths and 316,562 YLL avoided, if both power and industrial sectors met the ultra-low emission limits, accounting for 5.5% and 6.2% of the totals for the region, respectively. In order to improve regional air quality and to reduce human health risk effectively, coordinated control of various pollution sources should be implemented, and the





62 ultra-low emission control policy should be substantially expanded to industrial

63 boilers and other emission sources in non-power industries.

## 1. Introduction

65   Due to swift economic development and associated growth in demand for

66 electricity, coal-fired power plants have played an important role in energy

67 consumption and air pollutant emissions for a long time in China. For example, Zhao

68 et al. (2008) for the first time developed a "unit-based" emission inventory of primary

69 air pollutants from the coal-fired power sector in China and found that the sector

70 contributed 53% and 36% to the national total emissions of $SO_2$ and $NO_X$,

71 respectively, in 2005. Subsequently, $SO_2$ and $NO_X$ emissions from the power sector

72 were estimated to account respectively for 28%-53% and 29%-31% of the total annual

73 emissions in China during 2006-2010 according to the Multi-resolution Emission

74 Inventory for China (MEIC: http://www.meicmodel.org). To reduce high emissions

75 and improve air quality in China, advanced air pollutant control devices (APCDs)

76 have been gradually applied in the power sector including flue gas desulfurization

77 (FGD) for $SO_2$ control, selective catalytic reduction (SCR) for $NO_X$ control, and

78 high-efficiency dust collectors for primary particulate matter (PM) control. In recent

79 years, moreover, an "ultra-low emission" retrofitting policy has been widely

80 implemented, seeking to reduce the emission levels of coal-fired power plants to those

81 of gas-fired ones (i.e., 35, 50, and 5 mg/m$^3$ for $SO_2$, $NO_X$ and PM concentrations in

82 the flue gas). The expanded use of associated technologies has induced great changes

83 in the magnitude and spatio-temporal distribution of emissions from the power sector,

84 which have been analyzed and quantified by a series of studies (Y. Zhao et al., 2013;

85 Zhang et al., 2018; Liu et al., 2019; Tang et al., 2019; Y. Zhang et al., 2019). With the

86 updated unit-level information, for example, MEIC estimated that the power sector

87 shares of national total emissions declined from 28% to 22% and from 29% to 21%

88 for $SO_2$ and $NO_X$ during 2010-2015, respectively. Incorporating data from continuous

89 emission monitoring systems (CEMS), Tang et al. (2019) found that China's annual

90 power sector emissions of $SO_2$, $NO_X$ and PM declined by 65%, 60% and 72%

91 respectively during 2014-2017, due to the enhanced control measures. With a method

92 of collecting, examining and applying CEMS data, similarly, our previous work

93 indicated that the estimated emissions from the power sector would be 75%, 63% and



76% smaller than those calculated without CEMS data for $SO_2$, $NO_X$ and PM,
respectively (Y. Zhang et al., 2019).
Evaluations of emission estimates and the changed air quality from emission
abatement provide useful information on the sources of air pollution and the
effectiveness of pollution control measures. Air quality modeling is an important tool
for evaluating emission inventories, by comparing simulation results with available
observation data. Developed by the U.S. Environmental Protection Agency (USEPA),
the Models-3/Community Multi-scale Air Quality (CMAQ) system has been widely
used in China (Li et al., 2012; An et al., 2013; Wang et al., 2014; Han et al., 2015;
Zheng et al., 2017; Zhou et al., 2017; Chang et al., 2019). Han et al. (2015) conducted
CMAQ simulations with different emission inventories for East Asia, and found that
the simulated $NO_2$ columns using the emission inventory for the Intercontinental
Chemical Transport Experiment-Phase B (INTEX-B, Zhang et al., 2009) agreed better
with the satellite observations of the Ozone Monitoring Instrument (OMI) than the
simulations using the Regional Emission Inventory in Asia (REAS v1.11, Ohara et al.,
2007). Zhou et al. (2017) applied CMAQ to evaluate the national, regional and
provincial emission inventories for the Yangtze River Delta (YRD) region, and the
best model performance with the provincial inventory confirmed that the emission
estimate with more detailed information incorporated on individual power and
industrial plants helped improve the air quality simulation at relatively high horizontal
resolution. With air quality modeling, moreover, many studies have explored the
environment benefits of emission control measures taken in recent years (B. Zhao et
al., 2013; Huang et al., 2014; Li et al., 2015; Wang et al., 2015; Tan et al., 2017).
Wang et al. (2015) found that the implementation of the new Emission Standard of Air
Pollutants for Thermal Power Plants (GB13223-2011) could effectively reduce
pollutant emissions in China, and the environmental concentrations of $SO_2$, $NO_2$ and
$PM_{2.5}$ would decrease by 31.6%, 24.3% and 14.7% respectively in 2020 compared
with a baseline scenario for 2010. Li et al. (2015) found that the simulated
concentrations of $PM_{2.5}$ in the YRD region would decrease by 8.7%, 15.9% and 24.3%
from 2013 to 2017 in three scenarios with weak, moderate and strong emission
reduction assumptions in the Clean Air Action Plan, respectively.
Besides air quality itself, the health risk caused by air pollution exposures in
China is a major concern, especially to $PM_{2.5}$, a dominant pollutant in haze conditions.



Lim et al. (2012) has identified air pollution as a primary cause of global burden of
disease, especially in low- and middle-income countries, and PM$_{2.5}$ pollution was
ranked the fourth leading cause of death in China. Studies have shown that PM$_{2.5}$ is
closely related to several causes of death (Dockery et al., 1993; Hoek et al., 2013;
Lelieved et al., 2015; Butt et al., 2017; Gao et al., 2018; Maji et al., 2018; Hong et al.,
2019). For example, Lelieved et al. (2015) estimated that nearly 1.4 million people
died each year due to PM$_{2.5}$ exposure in China, 18% of which were related to the
emissions from the power sector. Based on simulated PM$_{2.5}$ using WRF-Chem and the
Integrated Exposure Response (IER) model, Gao et al. (2018) estimated that
emissions from the power sector results in 15 million years of life lost per year in
China. In addition to assessment of health risk based on observations of actual air
pollution levels, studies have also analyzed the health benefits of emission control
policies (Lei et al., 2015; Li and Li, 2018; Dai et al., 2019; Q. Zhang et al., 2019; X.
Zhang et al., 2019). Combining available observation and CMAQ modeling, Q. Zhang
et al. (2019) identified improved emission controls on industrial and residential
pollution sources as the main drivers of reductions in PM$_{2.5}$ concentrations from 2013
to 2017 in China, and estimated an annual reduction of PM$_{2.5}$-related deaths at 0.41
million. Lei et al. (2015) evaluated the health benefit of the Air Pollution Prevention
and Control Action Plan of China, and found that full realization of the air quality
goal in this plan could avoid 89 thousand premature deaths of urban residents, and
reduce 120,000 inpatient cases and 9.4 million outpatient service and emergency cases.
Focusing more regionally, X. Zhang et al. (2019) estimated the health impact of a
"coal-to-electricity" policy for residential energy use in the Beijing-Tianjin-Hebei
(BTH) region. They projected that the reduction in PM$_{2.5}$ concentrations from the
policy would avoid nearly 22,200 cases of premature death and 607,800 cases of
disease in the region in 2020. For areas with strong, industry-based economies, the
impact of air quality on public health can be more significant, attributed both to
relatively large and dense populations and to high pollution levels. Until now,
however, there have been few studies focusing on air quality improvement and
corresponding health benefits attributed to the implementation of the latest emission
control policies, notably China's ultra-low emission policy introduced above, at
regional scale.
As one of the most densely populated and economically developed regions, the


YRD region encompassing Shanghai and Anhui, Jiangsu, and Zhejiang provinces is a
key area for air pollution prevention and control in China (Huang et al., 2011; Li et al.,
2011; Li et al., 2012). It is also one of the regions with the earliest implementation of
the ultra-low emission policy on the power sector in the country. Quantification of
emission reductions and subsequent changes in air quality is crucial for full
understanding of the environmental benefits of the policy. To test the possible
improvement in the regional emission inventory, this study evaluated the air quality
modeling performance without and with CEMS data incorporated in the estimation of
emissions of the coal-fired power sector for the YRD region. The changes in regional
air quality and health risk resulting from the implementation of the ultra-low emission
policy for key industries were quantified combining the air quality modeling and the
health risk model. The results provide scientific support for incorporation of online
monitoring data to improve the estimation of air pollutant emissions, and for better
design of emission control policies based on their simulated environmental effects.

## 2. Methodology and data

### 2.1 Air quality modeling

In this study, we adopted CMAQ version 4.7.1 (UNC, 2010) to conduct air
quality simulations and to evaluate various emission inventories for the YRD region.
The model has performed well in Asia (Zhang et al., 2006; Uno et al., 2007; Fu et al.,
2008; Wang et al., 2009). Two one-way nested domains were adopted for the
simulations, and the horizontal resolutions were set at 27 and 9 km square grid cells
respectively, as shown in Figure 1. The mother domain (D1, 177 × 127 cells) covered
most of China and all or parts of surrounding countries in East, Southeast, and South
Asia. The second modeling region (D2, 118×121 cells) covered the YRD region,
including Jiangsu, Zhejiang, Shanghai, Anhui and parts of surrounding provinces.
Lambert Conformal Conic Projection was applied for the entire simulation area
centered at (110°E, 34°N) with two true latitudes, 40°N and 25°N. The simulated
periods were January, April, July and October 2015, as representative of the four
seasons. The first five days in each month were set as a spin-up period to provide
initial conditions for later simulations. The carbon bond gas-phase mechanism (CB05)



and AERO5 aerosol module were adopted in all the CMAQ modules, with details of
the model configuration found in Zhou et al. (2017). The initial concentrations and
boundary conditions for the D1 mother domain were the default clean profile, while
they were extracted from CMAQ outputs of D1 simulations for the nested D2 domain.
Normalized mean bias (NMB), normalized mean error (NME), and the correlation
coefficient (R) between the simulations and observations were selected to evaluate the
performance of CMAQ modeling (Yu et al., 2006). The hourly concentrations of $SO_2$,
$NO_2$, $O_3$ and $PM_{2.5}$ were observed at 230 state-operated ground stations of the national
monitoring network in the YRD region and were collected from Qingyue Open
Environmental Data Center (https://data.epmap.org).
The Weather Research and Forecasting (WRF) Model version 3.4
(http://www.wrf-model.org/index.php, Skamarock et al., 2008) was applied to provide
meteorological fields for CMAQ. Terrain and land-use data were taken from global
data of the U.S. Geological Survey (USGS), and the first-guess fields of
meteorological modeling were obtained from the final operational global analysis data
(ds083.2) by the National Center for Environmental Prediction (NCEP). Statistical
indicators including bias, index of agreement (IOA), and root mean squared error
(RMSE) were chosen to evaluate the performance of WRF modeling against
observations (Baker et al., 2004; Zhang et al., 2006). Ground observations at 3-h
intervals of four meteorological parameters including temperature at 2 m (T2),
relative humidity at 2 m (RH2), and wind speed and direction at 10 m (WS10 and
WD10) of 42 surface meteorological stations in the YRD region were downloaded
from the National Climatic Data Center (NCDC). The statistical indicators for WS10,
WD10, T2 and RH2 in the YRD region are summarized by month in Table S1 in the
Supplement. The discrepancies between WRF simulations and observations of these
meteorological parameters were generally acceptable (Emery et al., 2001). Better
agreements were found for T2 and RH2 with their biases ranging -0.62 to +0.12℃
and -3.20% to +6.60% respectively, and their IOAs were all within the benchmarks
(Emery et al., 2001). In general, WRF captured well the characteristics of main
meteorological conditions for the region.

**2.2 Emission inventories and cases**

The anthropogenic emissions from industry, residential and transportation sectors





for D1 and D2 were obtained from the national emission inventory developed in our
previous work (Xia et al., 2016). The total emissions excluding those of the power
sector of $SO_2$, $NO_X$ and PM for the YRD regions were estimated at 1501.0, 3468.4
and 2711.2 Gg for 2015, respectively. The emission inventory in Xia et al. (2016) was
developed using activity data at the provincial level, and the spatial distribution of
emissions by sector was conducted according to that of MEIC with the original spatial
resolution of $0.25° \times 0.25°$ in this study. The gridded emissions were further
downscaled to horizontal resolutions of 27 and 9 km in D1 and D2, respectively,
based on the spatial distribution of population (for residential sources), industrial
gross domestic product (for industrial sources), and the road network (for on-road
vehicles). The monthly variations of emissions from each sector were assumed to be
the same as in MEIC. In addition, the Model Emissions of Gases and Aerosols from
Nature developed under the Monitoring Atmospheric Composition and Climate
project (MEGAN-MACC, Guenther et al., 2012; Sindelarova et al., 2014) were
applied as the biogenic emission inventory, and the emissions of Cl, HCl and lightning
$NO_X$ were obtained from the Global Emissions Initiative (GEIA, Price et al., 1997).

For the power sector in the YRD region specifically, we adopted the unit-level

emission estimates from our previous study and allocated the emissions according to
the actual locations of individual units (Y. Zhang et al., 2019). As described in the
study, the detailed information at the power unit level was compiled based on official
environmental statistics including the installed capacity, fossil fuel consumption,
combustion technology, and APCDs. The geographic locations of power units were
taken initially from the environmental statistics, and then adjusted using Google Earth.
As shown in in Table 1, in total five emission cases were set for the air quality
simulation. Cases 1 and 2 used estimates of power sector emissions without/with
incorporation of CEMS data, and were compared against each other to evaluate the
benefit of online emission monitoring information in air quality simulation. Note Case
2 was set as the base case for further analysis of the effects of emission controls.
Based on the unit-level information from CEMS, Cases 3 and 4 calculated emissions
assuming that only power plant boilers, or both power plant and industrial boilers
meet the requirements of the ultra-low emission policy, respectively. The model
performances were compared with the base case to quantify the air quality
improvements that result from the policy. Case 5 removed all the emissions from the



power sector and thus helped to specify the contribution of the power sector to air
pollution in the YRD region.

The air pollutant emissions for all the cases are summarized by sector in Table S2

in the Supplement. With the CEMS data for the power sector incorporated, the total
emissions of $SO_2$, $NO_X$ and PM for the YRD region in Case 2 were estimated as 427,
618 and 331 Gg smaller than those in Case 1, with relative reductions of 20%, 14%
and 11% respectively. Benefiting from the implementation of the ultra-low emission
policy in the coal-fired power sector, the total emissions of anthropogenic $SO_2$, $NO_X$
and PM in Case 3 would further decline 123, 135 and 36 Gg compared to Case 2,
respectively. The analogous numbers for Case 4 were 1180, 1003, and 1315 Gg, and
the reduction rates compared to Case 2 were 70%, 27%, and 48% for $SO_2$, $NO_X$ and
PM, respectively. The implementation of the ultra-low emission policy for both power
and industry sectors would significantly reduce the primary pollutant emissions for
the YRD region. In Case 5 where the emissions from power sector were set as zero,
the total emissions of $SO_2$, $NO_X$ and PM were estimated to decrease by 11%, 7% and
2% respectively compared to Case 2.
**2.3 Health effect analysis**

We applied the IER model of the Global Burden of Disease Study (GBD) 2015

(Cohen et al., 2017) and quantified the impact of emission control policy on the
human health risk due to long-term exposure of $PM_{2.5}$ in the YRD region. The number
of attributable deaths and years of life lost (YLL) caused by long-term $PM_{2.5}$ exposure
for selected emission cases were calculated for five diseases in this study. It
considered the four adult diseases of the GBD study, including ischemic heart disease
(IHD), stroke (STK, including ischemic and hemorrhagic stroke), lung cancer (LC),
and chronic obstructive pulmonary disease (COPD), and a common disease among
young children, acute lower respiratory infection (LRI).

The health risks in the different emission cases were estimated following Gao et

al. (2018) with the updated information for 2015. First, the relative risk (RR) for each
disease was calculated using eq. (1):
$$RR_{i,j,k}(Cl) = \begin{cases} 1 + \partial_{i,j,k}(1 - e^{-\beta_{i,j,k}(Cl - C_0)^{\gamma_{i,j,k}}}), & Cl \geq C_0 \\ 1, & Cl < C_0 \end{cases} \tag{1}$$





where $i$, $j$, and $k$ represent the age, gender and disease type, respectively; $Cl$ is the
annual average $PM_{2.5}$ concentration simulated with WRF-CMAQ (the average of
January, April, July and October in this work); $C_0$ is the counterfactual concentration;
and $\partial$, $\beta$ and $\gamma$ are the parameters that describe the IER functions, as reported by
Cohen et al. (2017).

Secondly, the population attributable fractions (PAF) were calculated with RR
following eq. (2) by disease, age and gender subgroup:
$$PAF_{i,j,k} = \frac{RR_{i,j,k}(Cl)-1}{RR_{i,j,k}(Cl)} \tag{2}$$

Moreover, the mortality attributable to $PM_{2.5}$ exposure ($\triangle M$) was calculated
using eq. (3), where $y_0$ is the current age-gender-specific mortality rate, and $Pop$
represent the exposed population in the age-gender-specific group in grid cell $l$:
$$\triangle M_{i,j,k,l} = PAF_{i,j,k,l} \times y_{oi,j,k,l} \times Pop_{i,j,l} \tag{3}$$

The population data of the four provinces and cities in the YRD region were
obtained from statistical yearbooks (AHBS, 2016; JSBS, 2016; SHBS, 2016; ZJBS,
2016), and the gender distribution is shown in Table S3 in the Supplement. The
baseline age-gender-disease-specific mortality rates for the five diseases in China for
2015 were obtained from the Global Health Data Exchange database (GHDx,
https://vizhub.healthdata.org), as shown in Table S4 in the Supplement, and those by
province were calculated based on the provincial proportions in Xie et al. (2016). The
national population with the spatial resolution at 1×1 km in 2015 was provided by the
Landscan Global Demographic Dynamic Analysis Database developed by Oak Ridge
National Laboratory (ORNL) of the U.S. Department of Energy. As shown in Figure
S1 in the Supplement, the population densities in the YRD region are larger in
Shanghai, southern Jiangsu and northern Zhejiang.

Finally, the year of life lost (YLL) due to $PM_{2.5}$ exposure was calculated following
eq. (4), where $N$ represent the number of deaths in each age-gender-specific group,
and $L$ reflects the remaining life expectancy of the group:
$$YLL = \sum_{i,j} N_{i,j} \times L_{i,j} \tag{4}$$

The remaining life expectancies by age were obtained from the life tables from
the World Health Organization (WHO, https://www.who.int), as summarized in Table
S5 in the Supplement. The life expectancies at birth of Chinese males and females in
2015 were 74.8 and 77.7 years, respectively.





## 3. Results and discussion

### 3.1 Evaluation of emission estimates with air quality simulation

#### 3.1.1 Model performances without/with CEMS data

Air quality simulations based on emission inventories without/with incorporation of CEMS data for the coal-fired power sector (Cases 1 and 2, respectively) were conducted to test the improvement of emission estimates. Because of the combined influences of regional transport and chemical reactions of air pollutants in the atmosphere, nonlinear relationships were found between the changes of primary emissions and ambient concentrations of air pollutants. Compared to Case 1, the simulated annual average concentrations of $SO_2$, $NO_2$ and $PM_{2.5}$ in the YRD region were 10%, 7% and 6% lower respectively in Case 2, while that of $O_3$ was 7% higher, due to combined effects of emissions of volatile organic compounds (VOCs) and $NO_X$ precursors (Gao et al., 2005; Yang et al., 2012). Previous studies have shown that $O_3$ formation in most of the YRD region is under the "VOCs-limited" regime, i.e., the generation and removal of $O_3$ is more sensitive to VOCs and would be inhibited with high $NO_X$ concentrations in the atmosphere (Zhang et al., 2008; Liu et al., 2010; Wang et al., 2010; Xing et al., 2011). Therefore, the simulated reduced $NO_2$ concentrations from greater $NO_X$ emission control could elevate the $O_3$ concentration.

The simulated concentrations of $SO_2$, $NO_2$, $O_3$ and $PM_{2.5}$ based on the two emission inventories without/with CEMS data were compared with ground observations and are summarized in Table 2. Similar model performances were found for the two emission cases, with overestimation of $SO_2$, $NO_2$ and $PM_{2.5}$ and underestimation of $O_3$. The NMEs between the simulated and observed $SO_2$, $O_3$ and $PM_{2.5}$ concentrations were all smaller than 50% for both cases, and slightly worse simulation performances were found in July compared to the other three months. In particular, the correlation coefficients (R) between the simulated and observed $SO_2$ in July were only 0.17 and 0.14 for Cases 1 and 2, respectively, and the NMEs between the simulated and observed $NO_2$ were larger than 100%. In addition, greater overestimation of $SO_2$ and $PM_{2.5}$ by the model was found in July than other months, likely attributable to bias of WRF modeling. On one hand, the simulated WS10 in the YRD region in July (2.67 m/s) was slightly lower than the observation (2.75 m/s). The





underestimation in wind speed could weaken the horizontal diffusion and lead to
overestimation in air pollutant concentrations. Compared with the results from the
European Centre for Medium-range Weather Forecasts (ECMWF,
https://apps.ecmwf.int/datasets), on the other hand, the simulated boundary layer
height (BLH) was lower in WRF for all months. The NMBs of the WRF and ECMWF
BLH in January, April and October were around -15%, while that in July reached
-24%. The lower BLH would limit the vertical convection and diffusion of pollutants,
and thereby increase the surface concentrations of air pollutants. Similar to previous
studies (An et al., 2013; Liao et al., 2015; Tang et al., 2015; Gao et al., 2016; Wang et
al., 2016; Zhou et al., 2017), underestimation of $O_3$ was commonly found, and the
NMBs and NMEs between the simulation and observation for the two cases ranged
from -34.5% to 1.6% and from 27.5% to 37.1%, respectively. The underestimation in
$O_3$ likely resulted from bias in the estimation of precursor emissions. Suggested by the
positive NMBs of $NO_2$ modeling in Table 2, the $NO_X$ emissions were expected to be
overestimated in the two cases, even for Case 2 with the CEMS data incorporated
(which reflect the emission control benefits in recent years, as discussed in Y. Zhang
et al., 2019). In addition, underestimation of VOC emissions is likely due to
incomplete accounting of emission sources, particularly for uncontrolled or fugitive
leakage (Zhao et al, 2017). In a VOC-limited regime, therefore, the overestimation of
$NO_X$ and underestimation of VOCs would contribute to lower simulated $O_3$
concentrations than observations. In general, the simulations of both cases captured
well the temporal variations of $PM_{2.5}$ concentrations, with the R between the observed
and simulated concentrations around 0.9.
In general, better modeling performance in the YRD region was found in Case 2
than Case 1. The monthly average NMBs between the simulated and observed
concentrations of $SO_2$, $NO_2$, $O_3$ and $PM_{2.5}$ were -3.1%, 56.3%, -19.5% and -1.4% for
Case 2, smaller in absolute value than those for Case 1 at 8.2%, 68.9%, -24.6% and
7.6%, respectively. The bootstrap sampling (Gleser et al., 1996; He et al., 2017) was
further applied to test the significance of the improvements of Case 2 over Case 1. (A
significant difference is demonstrated if the confidence intervals of given statistical
indices sampled from the two cases do not overlap.) As can be seen in Table 2, the
modeling performances of the concerned species in Case 2 were improved
significantly in most instances compared to Case 1. For example, the improvement of



NMB for the $SO_2$ simulation was significant at the 99% confidence level for July and
October, and 95% for January. The improvement of NMB and NME for $NO_2$ was
significant at confidence levels of 99% and 95% respectively for April. The
improvement of NMB for $O_3$ was significant at the 95% confidence level for January,
and that of $PM_{2.5}$ at 95% for April and 99% for July. The statistical test confirms that
incorporation of online monitoring data in the emission inventory can improve the
regional air quality modeling for the YRD region
Figure 2 illustrates the spatial patterns of the simulated monthly $SO_2$, $NO_2$, $O_3$
and $PM_{2.5}$ concentrations for Case 2. For a given species, similar patterns were found
for different months. In general, the simulated concentrations of $SO_2$, $NO_2$ and $PM_{2.5}$
were larger in central and northern Anhui, southern Jiangsu, Shanghai and coastal
areas in Zhejiang, where large power and industrial plants are concentrated, as shown
in Figure S2 in the Supplement. In the highly populated cities (Shanghai, Nanjing,
Hangzhou, and Hefei; see their locations in Figure 1), the simulated concentrations of
pollutants were significantly larger than their surrounding areas. For example, the
simulated $SO_2$, $NO_2$ and $PM_{2.5}$ concentrations in Nanjing were 1.4, 1.3 and 1.2 times
of those in its nearby cities. The analogous numbers for Hangzhou were 2.5, 1.5 and
1.3. In contrast, the simulated $O_3$ concentrations were smaller in urban areas and
larger in suburban ones. For instance, the simulated $O_3$ in Nanjing, Shanghai, Hefei
and Hangzhou were 0.7, 0.4, 0.6 and 0.6 times of those in their surrounding areas,
respectively. The spatial distributions of the simulated $NO_2$ and $O_3$ concentrations in
Figure 2 also indicated that $O_3$ concentrations were less in the regions with higher
$NO_2$ concentrations, such as the megacity of Shanghai. The simulated high
concentrations of $NO_2$ in urban areas promotes titration of $O_3$, reducing its
concentrations. In addition, $O_3$ concentrations could remain relatively high after
transport from urban to the suburban areas due to relatively small emissions of $NO_X$
in the latter.
**3.1.2 Benefits of the "ultra-low" emission controls on air quality**
Table 3 summarizes the absolute and relative changes of the simulated monthly
concentrations of the concerned air pollutants in Cases 3-5 compared to the base case
(Case 2). The average contributions of the power sector to the total ambient
concentrations of $SO_2$, $NO_2$ and $PM_{2.5}$ for the four simulated months are estimated at



10.0%, 4.7%, and 2.3%, respectively, based on comparison of Cases 2 and 5. The
contributions to the concentrations were close to those of emissions at 10.7%, 6.6%,
and 1.6% for the three species (as indicated in Table S2), respectively. The larger
power sector contribution to the ambient $PM_{2.5}$ concentrations than to primary PM
emissions reflects high emissions of precursors of secondary sulfate and nitrate
aerosols. In general, limited contributions from the power sector were found for all
concerned species except $SO_2$, attributed to the gradually improved controls in the
sector. The further implementation of the ultra-low emission policy in the sector,
therefore, is expected to result in limited additional benefits for air quality. As shown
in Table 3, the absolute changes of the simulated $SO_2$, $NO_2$, $O_3$ and $PM_{2.5}$
concentrations in Case 3 compared to Case 2 were all smaller than 1 $\mu g/m^3$ for the
four months. Larger changes were found for primary pollutants ($SO_2$ and $NO_2$) than
those of secondary ones ($O_3$ and $PM_{2.5}$): the simulated monthly concentrations of $SO_2$
and $NO_2$ were 2.7%-6.1% and 2.0%-2.9% lower, while $PM_{2.5}$ was only 0.1%-1.3%
lower and $O_3$ 0.8%-2.2% higher, respectively. Much larger benefits were found when
the ultra-low emission policy was broadened from the power sector to the industrial
sector (Case 4), attributed to the dominant role of industry in air pollutant emissions in
the YRD region (Table S2). The simulated monthly concentrations of $SO_2$, $NO_2$ and
$PM_{2.5}$ were 1.5-2.0, 2.5-3.7, and 4.6-6.5 $\mu g/m^3$ lower compared to the base case,
respectively, or reduction rates of 32.9%-64.1%, 16.4%-22.8%, and 6.2%-21.6%. In
contrast, the simulated $O_3$ concentration was 0.8-4.8 $\mu g/m^3$ higher, with growth rates
ranging 2.6%-14.0%. As mentioned earlier, the YRD was identified as a VOC-limited
region, and reducing $NO_X$ emissions without any VOC controls would enhance $O_3$
concentrations. In order to alleviate regional air pollution including $O_3$, therefore,
coordinated controls of $NO_X$ and VOC emissions are urgently required. These would
include measures to reduce large sources of VOCs, notably in non-power industries
such as chemicals and refining and in solvent use (Zhao et al., 2017).
The relative changes in the simulated pollutant concentrations varied by month,
due to the combined influences of meteorology and secondary chemistry, and larger
changes were found for $SO_2$ and $PM_{2.5}$ in summer. As shown in Table 3, for example,
the average simulated $PM_{2.5}$ concentrations in July were 0.4 and 6.5 $\mu g/m^3$ lower
respectively under Cases 3 and 4 compared to Case 2, with the larger reduction than
other three months. This could result partly from the faster response of ambient





concentrations to the changed emissions of air pollutants with shorter lifetimes in
summer. Moreover, the formation of secondary pollutants like $PM_{2.5}$ would be
enhanced in summer, with more oxidative atmospheric conditions under high
temperature and strong sunlight.

Figures 3 and 4 illustrate the spatial distributions of the relative changes of
simulated pollutant concentrations in Cases 3 and 4 compared to Case 2, respectively.
As shown in Figure 3, the overall changes across the region due to ultra-low emission
controls in the power sector only were less than 10% for primary pollutants $SO_2$ and
$NO_2$, and 5% for secondary pollutants $PM_{2.5}$ and $O_3$. Larger changes in simulated $SO_2$
concentrations were found in central and northern Anhui as well as central and
southern Jiangsu, with relatively concentrated distribution of coal-fired power plants.
The changes of simulated $SO_2$ and $NO_2$ in Shanghai were tiny, due to few remaining
power plants subject to the ultra-low emission policy and thus few emission
reductions. Compared to Case 2, the $SO_2$ and $NO_X$ emissions in Case 3 were
estimated to be 2.2% and 0.8% lower respectively for Shanghai, much smaller than
for other provinces (6.1% and 2.5% for Anhui, 9.5% and 4.4% for Jiangsu, 5.5% and
2.7% for Zhejiang). The results suggest that the potential of emission reduction and
air quality improvement is limited from implementation of more stringent control
measures in the power sector alone, particularly in highly developed cities where air
pollution controls have already reached a relatively high level.

In Case 4, where both power plant and industrial boilers meet the ultra-low
emission requirement, the average reduction rates of simulated $SO_2$ and $NO_2$
concentrations compared to Case 2 were above 40% and 25% respectively for the
whole region, and the changes of secondary pollutants $O_3$ and $PM_{2.5}$ were also
significantly larger than those of Case 3 in most of the region. The relative changes of
$SO_2$ were found to be more significant than other species, as the $SO_2$ concentrations
are greatly affected by primary emissions. Due to the large number and wide
distribution of industrial plants throughout the YRD, moreover, there was little
regional disparity in the changed ambient $SO_2$ levels. In the central YRD, including
Shanghai, northern Zhejiang and southern Jiangsu, the changes in simulated $NO_2$ were
modest, resulting in clear enhancement of $O_3$ concentrations. The result suggests the
great challenges of $O_3$ pollution abatement in those developed areas, even with
aggressive measures on $NO_X$ control at power and industrial plants.





## 3.2 Evaluation of health benefits

### 3.2.1 PM$_{2.5}$ exposures in the YRD region

Figure 5 illustrates the spatial distributions of PM$_{2.5}$ concentrations for the base case (Case 2) and the differences of Cases 3 and 4 compared to the base case. The reduction of PM$_{2.5}$ concentrations from the implementation of the ultra-low emission policy in the power sector was less than 1 μg/m$^3$ over the YRD region (Figure 5b). Larger reductions (above 0.4 μg/m$^3$) were found in northern Anhui and northern and southern Jiangsu provinces, as those regions are the energy base of eastern China, with abundant coal mines and power plants with large installed capacities. With the policy expanded to industrial boilers, the simulated average PM$_{2.5}$ concentrations were 5.8 μg/m$^3$ lower for the whole region (Figure 5c). In particular, the difference was greater than 10 μg/m$^3$ along the Yangtze River, as there are many industrial parks located along the river containing a large number of big cement, iron & steel, and chemical industry plants. Stringent emission controls at those plants would result in significant benefits in air quality for local residents.

We further calculated the fractions of the population with different annual average PM$_{2.5}$ exposure levels in Cases 2-4, as shown in Figure 6. Compared to Case 2, slight differences in the population distribution by exposure level were found in Case 3, while the differences were much more significant in Case 4. The population fractions exposed to the average annual concentrations of PM$_{2.5}$ smaller than 35 μg/m$^3$, 35-45 μg/m$^3$ and 45-55 μg/m$^3$ were estimated to grow from 14% in Case 2 to 21% in Case 4, from 11% to 16%, and from 16% to 30%, respectively. Accordingly, the fraction exposed to PM$_{2.5}$ concentrations larger than 55 μg/m$^3$ declined from 59% to 33%. The implementation of ultra-low emission policy on both power plants and industry sectors thus proved an effective way in limiting the population exposed to high PM$_{2.5}$ levels.

### 3.2.2 Human health risk with base case emissions

The mortality and YLL caused by atmospheric PM$_{2.5}$ exposure with the base case emissions (Case 2) in the YRD region are shown in Table 4. The values in brackets represent the 95% confidence interval (CI) attributed to the uncertainty of IER curves

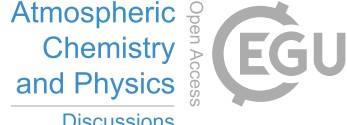

(i.e., uncertainties from other sources were excluded in the 95% CI estimation such as
air quality model mechanisms, emission inventories, and population data). With the
base case emissions, the NMB of the simulated and observed annual $PM_{2.5}$
concentrations (based on the four representative months) was calculated at -1.4% for
the YRD region. Therefore, the influence of the biases between the simulations and
observations on the estimated health risks was negligible and thus not considered in
this study. The total attributable deaths due to all diseases caused by $PM_{2.5}$ exposure in
the YRD region were estimated at 194,000 (114,000-282,000), with STK, IHD and
COPD causing the most deaths, accounting for 29%, 32% and 22% of the total
respectively. With larger populations in Anhui and Jiangsu (32% and 37% of the
regional total respectively), more deaths caused by $PM_{2.5}$ exposure were found in
these two provinces, at 34% and 41% of the total deaths respectively. Among all the
diseases, STK was found to cause the largest number of mortalities (19,600) in Anhui
with $PM_{2.5}$ exposure, IHD in Jiangsu (31,300), and COPD in Shanghai (4,400) and
Zhejiang (10,800). The total YLL caused by $PM_{2.5}$ exposure in the YRD region was
5.11 million years (3.16 - 7.18 million years). More YLL caused by $PM_{2.5}$ exposure
was found in Anhui and Jiangsu, accounting for 34% and 37% of the total in the YRD
region respectively. YLL caused by COPD were the largest in all the provinces, with
0.66, 0.19, 0.56 and 0.47 million years estimated for Anhui, Shanghai, Jiangsu and
Zhejiang, respectively. The spatial distribution of attributable deaths and YLL caused
by $PM_{2.5}$ exposure was basically consistent with that of population in the YRD region,
with correlation coefficients of 0.94 and 0.96 respectively. As shown in Figure 7,
higher health risks attributed to $PM_{2.5}$ pollution under the base case (Case 2) were
found in the areas along the Yangtze River, central Shanghai and some urban areas in
Anhui, all with higher population densities. We further compared the population
deaths attributable to $PM_{2.5}$ exposure calculated in this study with the reported total
deaths in provincial statistical yearbooks (AHBS, 2016; JSBS, 2016; SHBS, 2016;
ZJBS, 2016), and found that the deaths caused by $PM_{2.5}$ exposure accounted for 18%,
14%, 15% and 11% of the total deaths in Anhui, Jiangsu, Shanghai and Zhejiang
respectively for 2015. The numbers were larger than the estimate (6.9%) by Maji et al.
(2018), which focused on 161 cities in China.
Many studies have focused on the human health risks attributable to air pollution
in China, with considerable disparities between them due to different estimation





methods and health endpoints selected. Figure 8 compares the estimates of premature
deaths caused by $PM_{2.5}$ exposure in the YRD region in this and previous studies.
Relatively close results are found between studies for the same regions and periods.
For example, Hu et al. (2017) and Liu et al. (2016) estimated that the premature
deaths of adults (>30 years old) due to $PM_{2.5}$ exposure were 223,000 and 245,000
respectively in 2013 in the YRD region. However, the health endpoints in these two
studies were not completely consistent. COPD, LC, IHD and CEV (cerebrovascular
disease) were selected in Hu et al. (2017), while COPD, LC, IHD and STK were
chosen by Liu et al. (2016). The deaths caused by $PM_{2.5}$ exposure in Shanghai were
estimated at 19,000, 15,000, and 16,000 respectively in Maji et al. (2018), Song et al.
(2017) and this study, respectively. The IER model and the same health endpoints
were adopted in all three studies, while the $PM_{2.5}$ concentrations were derived from
ground observations in the former two studies instead of air quality simulation in this
study. The premature deaths attributable to $PM_{2.5}$ exposure in the YRD region in 2015
were estimated at 122,000 in Maji et al. (2018) and 194,000 in this study respectively,
with the discrepancy resulting in part from inclusion of only typical cities instead of
all cities of the YRD region in the estimation of the former. There are clear disparities
in estimates of premature deaths for different years. For example, the death estimates
caused by $PM_{2.5}$ exposure in 2015 were generally smaller than those in 2013. As the
population and age distributions remained relatively stable over the two years (AHBS,
2016; JSBS, 2016; SHBS, 2016; ZJBS, 2016), the reduced estimated premature
deaths result to some extent from emission abatement and air quality improvement.
According to relevant studies of Shanghai in particular (Lelieveld et al., 2013; 2015;
Liu et al., 2016; Xie et al., 2016; Hu et al., 2017; Song et al., 2017; Maji et al., 2018),
the premature deaths attributable to $PM_{2.5}$ exposure increased from 2005 to 2013 and
then declined afterwards, reflecting the health benefit of air pollution control
measures in Shanghai in recent years.

### 3.2.3 Benefits of emission controls on human health

Tables 5 and 6 respectively summarize the avoided premature deaths and YLL by
disease and region that would result from implementation of the ultra-low emission
control policy and thereby reduced $PM_{2.5}$ pollution in the YRD region. If only the
coal-fired power sector meet the ultra-low emission limits (Case 3), nearly 305





premature deaths would be avoided compared to the base case emissions in 2015,
with a tiny reduction rate of only 0.16%. If the policy is strictly implemented for the
industrial sector as well (Case 4), 10,651 premature deaths could be avoided with a
reduction rate at 5.50%. The largest numbers of avoided premature deaths were found
in Anhui and Jiangsu, accounting collectively for 88.2% and 68.7% of the total
avoided deaths in Cases 3 and 4 respectively. The greatest impacts from reduced
PM$_{2.5}$ concentrations were found for STK, of which the avoided deaths were
calculated at 85 and 2848 in Cases 3 and 4, respectively. The health effects of
emission control policies in the YRD region have been investigated in previous
studies. Using the IER model, Dai et al. (2019) chose the premature deaths from IHD,
CEV, COPD and LC as health endpoints, and found that the Clean Air Action Plan
would avoid 3,439 deaths caused by PM$_{2.5}$ exposure in Shanghai, more than those in
both Case 3 and Case 4 in this study (5 and 1,185 respectively). Applying
environmental health risk and valuation methods, Li and Li (2018) found that 15,709
premature deaths attributable to air pollution could be avoided in 2015 if the PM$_{2.5}$
concentrations in Jiangsu province were assumed to meet the National Ambient Air
Quality Standard (GB3095-2012, 35 µg/m$^3$ as the annual average). The estimate is
much more than those calculated in Case 3 and Case 4 (177 and 4,114 deaths
respectively). The larger health benefits estimated in those two studies result from
their assumption of emission control measures covering a much wider range of sectors
including energy, industry, transportation, construction, and agriculture, while only the
ultra-low emission policy was assumed for the power and industry sectors in this
study. The comparisons illustrate that the health benefits from emission control in the
power sector alone is limited, and that controls in other sectors are essential. In
addition, the different methods and inconsistent data sources partly led to the
discrepancies. For the particle exposure estimation, as an example, Dai et al. (2019)
adopted the BENMAP-CE model (Environmental Benefits Mapping and Analysis
Program-Community Edition, Yang et al. (2013)) to simulate the ambient PM$_{2.5}$
concentrations, while Li and Li (2018) used the average of monitored PM$_{2.5}$
concentrations. As shown in Table 6, the avoided YLL for Case 3 and Case 4 were
estimated at 8744 and 316,562 years respectively compared to the base case,
confirming again the greatly improved health benefits from implementation of
ultra-low emission policy for the industry sector in addition to the power sector. The



largest avoided YLL were found in Anhui and Jiangsu in the YRD region, accounting
collectively for 86% and 65% of the total avoided YLL in Cases 3 and 4 respectively.
Comparing Case 3 to Case 4, the fractions of both avoided deaths and YLL were
clearly higher for Shanghai and part of Zhejiang, implying a greater health benefit of
emission controls at industry sources in these relatively industrialized urban regions.
The reduced $PM_{2.5}$ concentrations led to the largest avoided YLL of COPD in both
cases (3,118 and 119,300 years in Cases 3 and 4, respectively).
Figure 9 illustrates the spatial distributions of the avoided deaths and YLL from
the ultra-low emission policy in the YRD region. When the policy was implemented
only for coal-fired power plants, the health benefits were small and the regional
differences relatively insignificant, with the avoided deaths and YLL smaller than 10
persons and 100 years respectively for all of the grid cells (Figure 9a and 9b). When
the policy was implemented both in power and industry sectors, more avoided deaths
(>40 person/grid cell) and YLL (>400 years/grid cell) were found in northern Anhui,
southern Jiangsu, central Shanghai and northern Zhejiang (Figure 9c and 9d). The
spatial correlation coefficient between the avoided YLL in Case 4 and population was
0.93, indicating that the implementation of the emission control policy would lead to
greater health benefits for areas with intensive economic activity and dense
populations.

## 4. Conclusions

We evaluated the improvement of emission estimation by incorporating CEMS
data for the power sector, and explored the air quality and health benefits from the
ultra-low emission control policy for the YRD region through air quality modeling. In
general, the bias between ground observations and simulations based on the emission
inventory with CEMS data incorporated was smaller than that without, suggesting that
appropriate use of online monitoring information helped improve the emission
estimation and model performance. Compared to the base case in which CEMS data
were incorporated in emission estimation, the simulated monthly concentrations of all
the concerned species ($SO_2$, $NO_2$, $O_3$, and $PM_{2.5}$) differed less than 7% when the
ultra-low emission policy was enacted only in the coal-fired power sector, given its
small fraction of total emissions. When the policy was implemented for the industrial





sector as well, larger differences in air quality from the base case were found, with the
simulated concentrations of $SO_2$, $NO_2$ and $PM_{2.5}$ respectively 33%-64%, 16%-23%
and 6%-22% lower and $O_3$ 3%-14% higher, depending on the month.
Nearly 305 premature deaths and 8,744 years of YLL would be avoided if the
policy was implemented for the power sector alone, and benefits would reach 10,651
premature deaths and 316,562 YLL avoided with the policy enacted for both power
and industry sectors. The study revealed the limited potential for further emission
reduction and air quality improvement via controls in the power sector alone. Along
with stringent emission control in that sector, the coordinated control of emissions
from non-power industrial sources would be essential to effectively improve air
quality and reduce associated human health risks. Moreover, more attention needs to
be paid to control of VOC to limit $O_3$ formation resulting from reduction of $NO_X$ in
the region.

## Data availability


All data in this study are available from the authors upon request.

## Author contributions


YZhang developed the strategy and methodology of the work and wrote the draft.
YZhao improved the methodology and revised the manuscript. MG provided useful
comments on the health risk analysis. XB provided emission monitoring data. CPN
revised the manuscript.

## Competing interests


The authors declare that they have no conflict of interest.

## Acknowledgments


This work was sponsored by the Natural Science Foundation of China (41922052 and
91644220), National Key Research and Development Program of China
(2017YFC0210106), and a Harvard Global Institute award to the Harvard-China





667 Project on Energy, Economy and Environment. We would also like to thank Tsinghua

668 University for the free use of national emissions data (MEIC).

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





**Figure captions**

Figure 1. The modeling domain and the locations of the concerned provinces and their capital cities. The numbers 1-4 represent the cities of Nanjing, Hefei, Shanghai and Hangzhou, respectively. The map data provided by Resource and Environment Data Cloud Platform are freely available for academic use (http://www.resdc.cn/data.aspx?DATAID=201), © Institute of Geographic Sciences & Natural Resources Research, Chinese Academy of Sciences.

Figure 2. The spatial distributions of the simulated monthly $SO_2$, $NO_2$, $O_3$ and $PM_{2.5}$ concentrations for Case 2 in D2 (unit: $\mu g/m^3$).

Figure 3. The spatial distributions of the relative changes (%) in the simulated monthly $SO_2$, $NO_2$, $O_3$ and $PM_{2.5}$ concentrations between Cases 2 and 3 (Case 2-Case 3) in D2.

Figure 4. The spatial distributions of the relative changes (%) in the simulated monthly $SO_2$, $NO_2$, $O_3$ and $PM_{2.5}$ concentrations between Cases 2 and 4 (Case 2-Case 4) in D2.

Figure 5. The spatial distributions of the annual $PM_{2.5}$ concentrations (average of January, April, July and October) for Case 2 (a) and the reduced annual $PM_{2.5}$ concentrations for Cases 3 (b) and 4 (c) in the YRD region (unit: $\mu g/m^3$). Note the different color ranges in the panels for easier visualization.

Figure 6. The population fractions exposed to different levels of $PM_{2.5}$ in the YRD region for Cases 2 (a), 3 (b), and 4 (c).

Figure 7. The spatial distributions of the mortality (a) and YLL (b) attributable to $PM_{2.5}$ exposure in Case 2 at a horizontal resolution of 9 km.

Figure 8. Comparisons of the estimated mortality attributable to $PM_{2.5}$ exposure in various studies for the YRD region.

Figure 9. The spatial distributions of the avoided deaths and YLL attributable to the reduced $PM_{2.5}$ exposure with ultra-low emission policy implementation at a horizontal resolution of 9 km. Note the different color ranges in the panels for easier visualization.





# Tables

**Table 1 Descriptions of the emission cases.**

| Case | Description |
| --- | --- |
| Case 1 | The emissions of coal-fired power sector were estimated based on the emission factor method without CEMS data incorporated. |
| Case 2 | The emissions of coal-fired power sector were estimated based on the improved method by Y. Zhang et al. (2019), with CEMS data incorporated. |
| Case 3 | All the coal-fired power plants in the YRD region were assumed to meet the requirement of the ultra-low emission policy. |
| Case 4 | All the coal-fired power and industrial boilers in the YRD region were assumed to meet the requirement of the ultra-low emission policy. |
| Case 5 | The emissions of all coal-fired power plants were set at zero. |


**Table 2 Comparisons of the observed and simulated monthly SO$_2$, NO$_2$, O$_3$ and PM$_{2.5}$ concentrations in Cases 1 and 2 in the YRD region.**

| Pollutant | | R | | NMB (%) | | NME (%) | |
|---|---|---|---|---|---|---|---|
| | | Case 1 | Case 2 | Case 1 | Case 2 | Case 1 | Case 2 |
| SO$_2$ | Jan | 0.72 | 0.89↑ | 11.44 | 0.52↑** | 26.83 | 24.22↑ |
| | Apr | 0.36 | 0.45↑ | -18.45 | -22.62 | 31.65 | 34.81 |
| | Jul | 0.17 | 0.14 | 36.84 | 15.72↑*** | 58.69 | 48.44↑ |
| | Oct | 0.59 | 0.57 | 14.59 | 1.15↑*** | 32.49 | 29.22↑* |
| NO$_2$ | Jan | 0.72 | 0.73↑ | 42.74 | 34.92↑* | 44.25 | 37.88↑ |
| | Apr | 0.64 | 0.69↑ | 69.24 | 48.72↑*** | 70.24 | 51.81↑** |
| | Jul | 0.71 | 0.71 | 145.42 | 131.65↑* | 145.42 | 131.65↑* |
| | Oct | 0.70 | 0.69 | 58.15 | 47.73↑* | 58.86 | 49.41↑* |
| O$_3$ | Jan | 0.74 | 0.75↑ | -16.90 | -6.40↑** | 30.53 | 28.60↑ |
| | Apr | 0.78 | 0.67 | -14.88 | -9.89↑ | 23.14 | 27.48 |
| | Jul | 0.78 | 0.79↑ | -34.49 | -28.46↑ | 37.11 | 32.77↑ |
| | Oct | 0.80 | 0.78 | -30.37 | -28.28↑ | 34.32 | 33.60↑ |
| PM$_{2.5}$ | Jan | 0.89 | 0.90↑ | -0.28 | 1.63 | 16.27 | 15.21↑ |
| | Apr | 0.76 | 0.76 | 9.94 | 2.57↑** | 21.30 | 19.26↑ |
| | Jul | 0.64 | 0.63 | 30.44 | 24.08↑*** | 37.66 | 34.29↑* |
| | Oct | 0.75 | 0.75 | 5.40 | -11.80 | 23.34 | 22.28 |

Note: The arrow represents that the simulation results in Case 2 were improved compared to Case 1. *, **, and *** indicate the improvements are statistically significant with confidence levels of 90%, 95%, and 99 %, respectively. The R, NMB and NME were calculated using the following equations ($P$, $O$, $\bar{P}$, and $\bar{O}$ represent the simulation, observation, averaged simulation and averaged observation values, respectively):

$$NMB = \frac{\sum_{i=1}^{n}(P_i - O_i)}{\sum_{i=1}^{n} O_i} \times 100\% \qquad ; \qquad NME = \frac{\sum_{i=1}^{n}|P_i - O_i|}{\sum_{i=1}^{n} O_i} \times 100\% \qquad ;$$

$$R = \frac{\sum_{i=1}^{n}(P_i - \bar{P})(O_i - \bar{O})}{\sqrt{\sum_{i=1}^{n}(P_i - \bar{P})^2 \sum_{i=1}^{n}(O_i - \bar{O})^2}}$$





Atmospheric Chemistry and Physics Discussions Open Access

**Table 3 The relative (%) and absolute changes ($\mu g/m^3$, in parentheses) of the simulated monthly pollutant concentrations in different cases relative to Case 2 in the YRD region.**

| Pollutant | (Case 3 - Case 2) / Case 2 | | | | (Case 4 - Case 2) / Case 2 | | | | (Case 5 - Case 2) / Case 2 | | | |
|---|---|---|---|---|---|---|---|---|---|---|---|---|
| | Jan | Apr | Jul | Oct | Jan | Apr | Jul | Oct | Jan | Apr | Jul | Oct |
| $SO_2$ | -2.7 | -4.8 | -6.1 | -4.3 | -32.9 | -57.3 | -64.1 | -55.1 | -4.3 | -11.4 | -12.1 | -12.1 |
| | (-0.2) | (-0.2) | (-0.1) | (-0.2) | (-2.0) | (-1.8) | (-1.5) | (-2.4) | (-0.3) | (-0.4) | (-0.3) | (-0.5) |
| $NO_2$ | -2.0 | -2.9 | -2.0 | -2.5 | -16.4 | -21.9 | -17.1 | -22.8 | -2.6 | -5.9 | -4.1 | -6.2 |
| | (-0.4) | (-0.4) | (-0.3) | (-0.4) | (-3.2) | (-3.0) | (-2.5) | (-3.7) | (-0.5) | (-0.8) | (-0.6) | (-1.0) |
| $O_3$ | 1.7 | 2.2 | 0.8 | 2.2 | 10.4 | 9.7 | 2.6 | 14.0 | -2.0 | 2.7 | -1.6 | 4.5 |
| | (0.4) | (0.9) | (0.3) | (0.8) | (2.6) | (4.1) | (0.8) | (4.8) | (-0.5) | (1.2) | (-0.5) | (1.5) |
| $PM_{2.5}$ | -0.1 | -0.5 | -1.3 | -0.5 | -6.2 | -14.6 | -21.6 | -14.3 | -1.7 | -2.4 | -4.3 | -0.9 |
| | (-0.1) | (-0.2) | (-0.4) | (-0.2) | (-4.6) | (-6.0) | (-6.5) | (-6.3) | (-1.3) | (-1.0) | (-1.3) | (-0.4) |





**Table 4 The estimated mortality and YLL attributable to PM$_{2.5}$ exposures in Case 2 over the YRD region.**

| | STK | IHD | COPD | LC | LRI | Total |
|---|---|---|---|---|---|---|
| Deaths (×10³ person) | | | | | | |
| Anhui | 19.6 (10.7-29.0) | 19.1 (11.0-29.8) | 15.2 (9.8-21.0) | 8.0 (5.5-10.3) | 3.1 (2.4-3.8) | 65.0 (39.4-93.9) |
| Shanghai | 4.3 (2.3-6.5) | 4.2 (2.4-6.6) | 4.4 (2.7-6.1) | 2.6 (1.7-3.3) | 0.8 (0.6-1.0) | 16.3 (9.8-23.4) |
| Jiangsu | 23.6 (12.7-35.0) | 31.3 (17.8-48.8) | 12.8 (8.1-17.7) | 8.1 (5.5-10.5) | 3.7 (2.8-4.5) | 79.5 (46.8-116.5) |
| Zhejiang | 8.7 (4.2-13.4) | 6.8 (3.6-10.4) | 10.8 (6.2-15.4) | 5.0 (3.1-6.9) | 1.6 (1.1-2.0) | 32.9 (18.2-48.2) |
| YRD | 56.2 (29.9-83.8) | 61.4 (34.7-95.5) | 43.3 (26.8-60.2) | 23.6 (15.8-31.0) | 9.2 (7.0-11.3) | 193.8 (114.2-281.9) |
| YLL (×10⁴ year) | | | | | | |
| Anhui | 30.1 (16.6-44.0) | 29.6 (17.3-45.6) | 66.0 (42.3-91.1) | 34.5 (23.7-44.4) | 13.6 (10.4-16.4) | 173.7 (110.3-241.5) |
| Shanghai | 6.7 (3.6-9.8) | 6.5 (3.8-10.0) | 19.0 (11.9-26.2) | 11.0 (7.4-14.4) | 3.5 (2.7-4.3) | 46.7 (29.4-64.8) |
| Jiangsu | 36.2 (19.7-53.1) | 48.6 (28.0-74.7) | 55.6 (35.0-76.7) | 35.0 (23.6-45.6) | 16.0 (12.3-19.4) | 191.4 (118.5-269.5) |
| Zhejiang | 13.3 (6.5-20.5) | 10.6 (5.7-16.0) | 46.9 (26.7-66.6) | 21.8 (13.6-30.0) | 6.8 (4.8-8.9) | 99.4 (57.2-141.9) |
| YRD | 86.3 (46.3-127.4) | 95.3 (54.7-146.4) | 187.4 (115.9-260.6) | 102.3 (68.3-.134.4) | 40.0 (30.1-48.9) | 511.3 (315.5-717.7) |


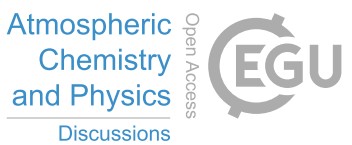

**Table 5 The reduced attributable deaths (person) and rates (in parentheses) resulting from implementation of the ultra-low emission policy in the YRD region.**

| | STK | IHD | COPD | LC | LRI | Total |
|---|---|---|---|---|---|---|
| **Case 3** | | | | | | |
| Anhui | 26 (0.13%) | 19 (0.10%) | 24 (0.16%) | 18 (0.22%) | 6 (0.18%) | 92 (0.14%) |
| Shanghai | 1 (0.03%) | 1 (0.02%) | 1 (0.03%) | 1 (0.04%) | 0 (0.04%) | 5 (0.03%) |
| Jiangsu | 51 (0.22%) | 51 (0.16%) | 34 (0.27%) | 30 (0.37%) | 11 (0.31%) | 177 (0.22%) |
| Zhejiang | 7 (0.08%) | 4 (0.06%) | 11 (0.10%) | 7 (0.14%) | 2 (0.13%) | 31 (0.10%) |
| YRD | 85 (0.15%) | 74 (0.12%) | 71 (0.16%) | 55 (0.23%) | 19 (0.21%) | 305 (0.16%) |
| **Case 4** | | | | | | |
| Anhui | 901 (4.59%) | 650 (3.41%) | 848 (5.56%) | 605 (7.60%) | 196 (6.23%) | 3200 (4.92%) |
| Shanghai | 281 (6.46%) | 204 (4.84%) | 348 (7.95%) | 277 (10.86%) | 75 (9.20%) | 1185 (7.26%) |
| Jiangsu | 1192 (5.05%) | 1179 (3.76%) | 794 (6.19%) | 684 (8.47%) | 264 (7.14%) | 4114 (5.17%) |
| Zhejiang | 475 (5.49%) | 283 (4.16%) | 765 (7.06%) | 491 (9.77%) | 138 (8.72%) | 2152 (6.54%) |
| YRD | 2848 (5.06%) | 2316 (3.77%) | 2755 (6.37%) | 2058 (8.71%) | 673 (7.28%) | 10651 (5.50%) |





**Table 6 The reduced cases and rates (in parentheses) of YLL resulting from implementation of the ultra-low emission policy in the YRD region.**

|  | STK | IHD | COPD | LC | LRI | Total |
|---|---|---|---|---|---|---|
| **Case 3** | | | | | | |
| Anhui | 396 (0.13%) | 285 (0.10%) | 1058 (0.16%) | 760 (0.22%) | 243 (0.18%) | 2743 (0.16%) |
| Shanghai | 17 (0.03%) | 13 (0.02%) | 60 (0.03%) | 45 (0.04%) | 13 (0.04%) | 148 (0.03%) |
| Jiangsu | 783 (0.22%) | 774 (0.16%) | 1480 (0.27%) | 1282 (0.37%) | 491 (0.31%) | 4809 (0.25%) |
| Zhejiang | 107 (0.08%) | 66 (0.06%) | 483 (0.10%) | 301 (0.14%) | 87 (0.13%) | 1044 (0.11%) |
| YRD | 1303 (0.15%) | 1138 (0.12%) | 3118 (0.16%) | 2388 (0.23%) | 834 (0.21%) | 8744 (0.17%) |
| **Case 4** | | | | | | |
| Anhui | 13733 (4.56%) | 9946 (3.36%) | 36709 (5.56%) | 26218 (7.60%) | 8480 (6.23%) | 95086 (5.47%) |
| Shanghai | 4284 (6.43%) | 3127 (4.78%) | 15083 (7.95%) | 11993 (10.86%) | 3233 (9.20%) | 37719 (8.07%) |
| Jiangsu | 18192 (5.02%) | 18066 (3.72%) | 34393 (6.19%) | 29638 (8.47%) | 11451 (7.14%) | 111740 (5.84%) |
| Zhejiang | 7297 (5.49%) | 4380 (4.13%) | 33115 (7.06%) | 21255 (9.77%) | 5972 (8.72%) | 72018 (7.25%) |
| YRD | 43506 (5.04%) | 35518 (3.73%) | 119300 (6.37%) | 89104 (8.71%) | 29135 (7.28%) | 316562 (6.19%) |





**Figure 1**

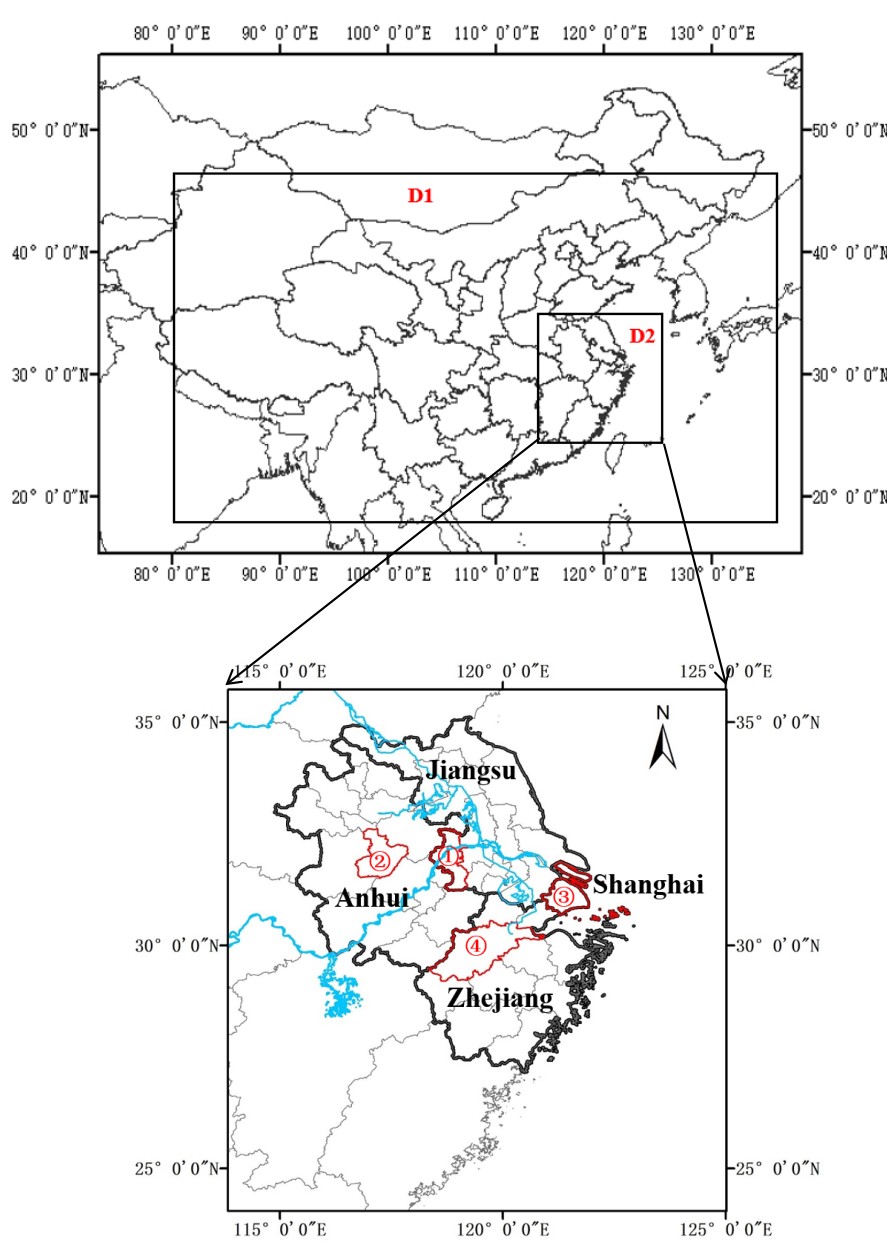





Figure 2



**Figure 3**

**Figure 4**





**Figure 5**

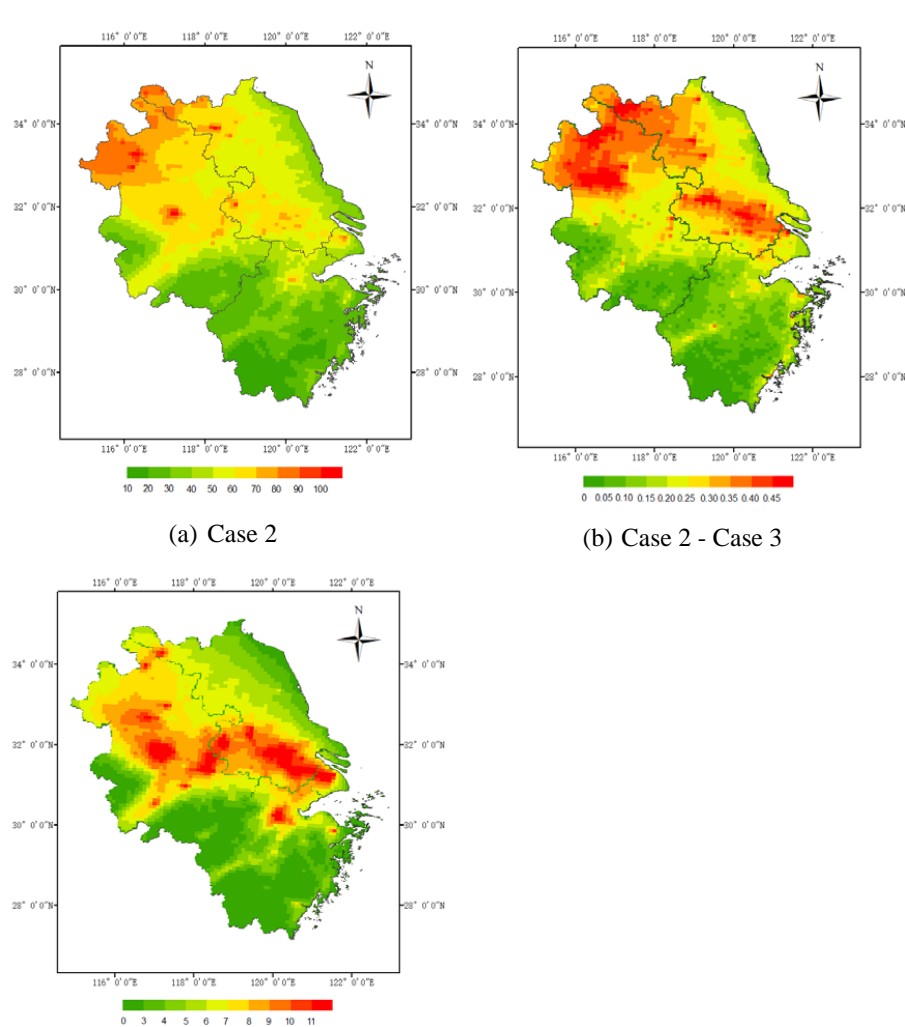

(a) Case 2

(b) Case 2 - Case 3

(c) Case 2 - Case 4





**Figure 6**

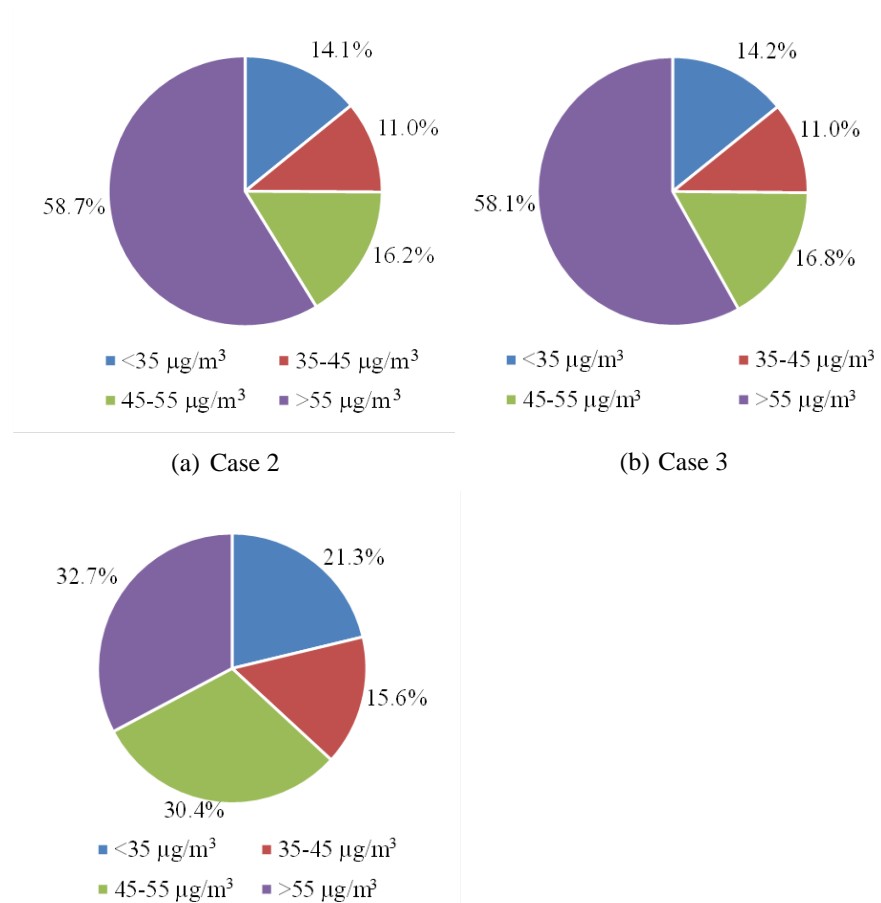



**Figure 7**

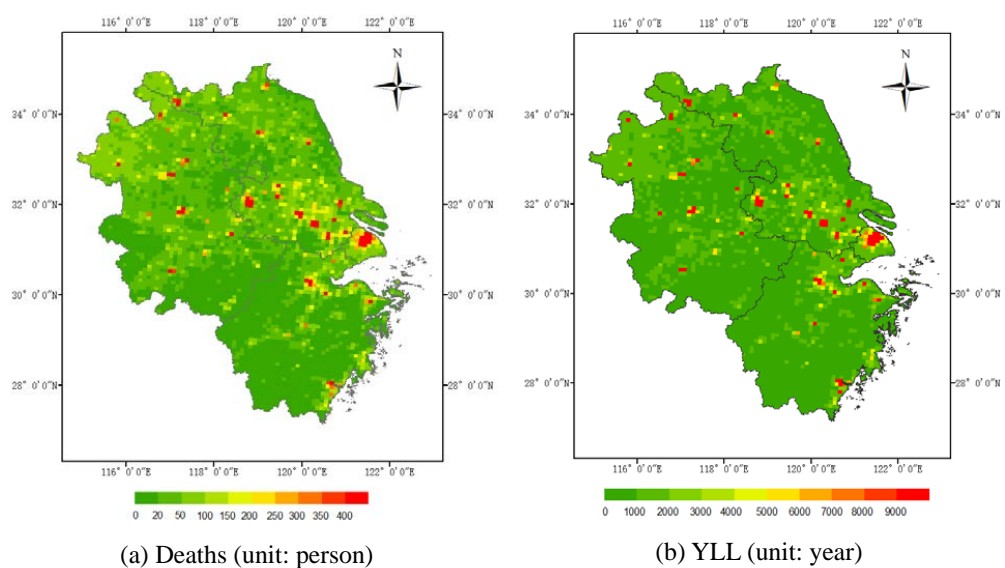

(a) Deaths (unit: person)        (b) YLL (unit: year)



**Figure 8**

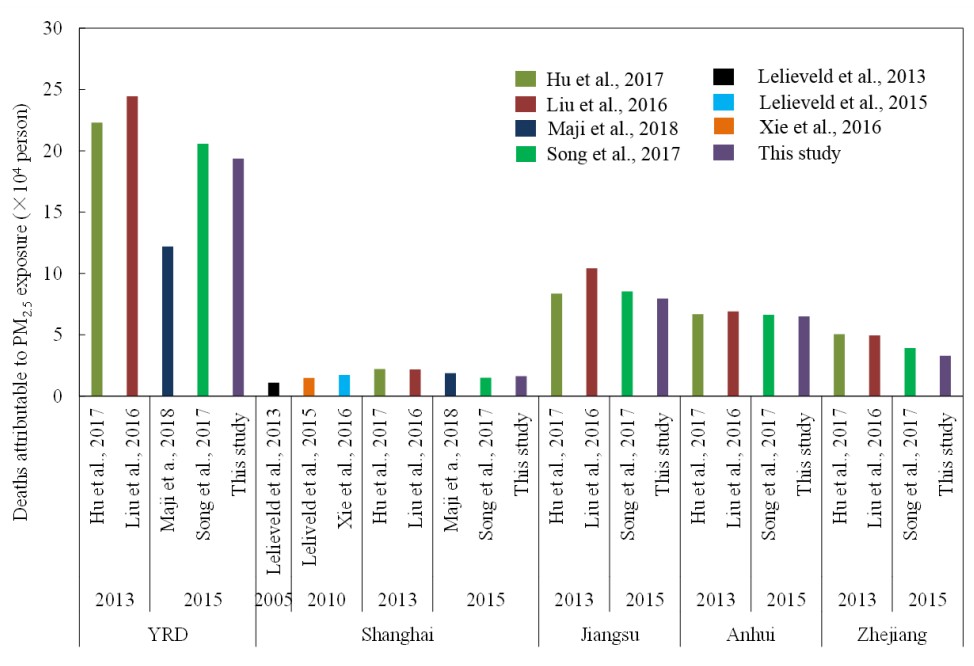



**Figure 9**



(a) Avoided deaths in Case 3 (unit: person)

(b) Avoided YLL in Case 3 (unit: year)

(c) Avoided deaths in Case 4 (unit: person)

(d) Avoided YLL in Case 4 (unit: year)