# Peer review of "Air quality and health benefits from ultra-low emission control policy indicated by continuous emission monitoring: A case study in the Yangtze River Delta region, China Yan Zhang1,2, Yu Zhao1,3\*, Meng Gao4, Xin Bo5, Chris P. Nielsen6 1. State"

_Atmospheric Chemistry and Physics, 2020_

## Referee Comment (RC1) · Anonymous Referee #2 · 10 Dec 2020

Under the big pressure of air quality improvement, China has been conducting a series of measures to reduce the emissions of air pollutants, including the "ultra-low" emission policy for power and specific industrial sectors. Continuous emission monitoring system (CEMS) has been gradually installed and operated to examine the real emission status of individual plants. Besides, CEMS provided opportunities of improving the understanding of air pollutant emissions for atmospheric science community. Focusing on the YRD, one of the most economically developed region in China, this paper presented an extended study based on the previously developed emission inventory of

power sector using the CEMS measurement (Y. Zhang et al., 2019). The authors applied air quality model and evaluated the YRD emission inventory with CEMS incorporated. They further combined the air quality model and exposure-response model, and analyzed the benefit of ultra-low emission policy on air quality and health. It provides the evidence of emission data improvement and the environmental health implication of current air pollution control policy, thus it fits the scope of Atmos Chem Phys. In general the paper is well organized and clearly written. I would suggest its acceptance for publication, with minor revisions or discussions conducted on the following issues.

1. Abstract. Some sentences are unclear and should be rewritten. Lines 49-50: "11%, 7% and 2% of SO2, NOX and PM" for which case?

2. Lines 155-157, did Gao et al. (2018) (and some other studies maybe) include the CEMS data for emission inventory development and stress the ultra-low emission policy?

3. Lines 226-234. It seems that you apply the inventory by Xia et al. (2016) but the spatiotemporal distribution of MEIC? Why not use MEIC directly?

4. Lines 251-253. Some information is missing here. Case 4 assumes both power and industrial boilers would meet the ultra-low emission limit. Do the two sectors share the same limit? In your previous work (Y. Zhang et al., 2019) you analyzed the emission limit for power sector with CEMS incorporated, but how about references for industrial boilers? Description should be given here.

5. Table 1 seems unnecessary, while the emission data in difference cases are more important (Table S2). Could you combine Table 1 and Table S2?

6. Line 326-328. The difference for SO2 was 10%, did this contradict the statement in lines 45-46 in the abstract: "... SO2, NO2, O3 and PM2.5 concentrations compared to those of Case 2, our base case for policy comparisons, were less than 7% for all pollutants"?

7. Lines 359-360. I cannot quite understand the numbers. You are comparing NMBs and NMEs for two cases, then what did the two groups of numbers exactly stand for?

8. Line 367. Is it commonly known that the YRD is under VOC-limited regime in terms of O3 formation? At least some literatures should be provided here.

9. Lines 435-437. Did the "ultra-low" emission policy include the limit of VOCs? Or did CEMS include the information of VOCs? Some explanations should be given here.

10. Lines 539-540: Any reason could be provided?

11. Language issues.

Line 119: "environmental concentrations"? Line 125: delete the word "itself"; Line 504: change "sectors" to "boilers"; Lines 531-534: rewrite the sentence

---

## Referee Comment (RC2) · Anonymous Referee #3 · 6 Feb 2021

This paper evaluated the potential benefit of the ultra-low emission policy on both air quality and human health in the YRD region. No novel technique was developed, or new scientific finding was reported. The results can still provide some scientific reference for related emission control policy and health burden caused by air pollution over the YRD region. Overall, this paper is well written, but more description in the methodology is still needed. A major revision is suggested, and my specific comments are listed as follows.

Specific comments: 1. Line 55, in the abstract section, "874 years", but according to

[Figure]

Table6, it should be "8744 years of life loss".

2. In the methodology section, please generally introduce the method of how to incorporate the CEMS data and cite the references which have the detailed description.

3. Why still using the old version of the CMAQ model? The current CMAQ model (v5.2 or v5.3) has incorporated several trace gas chemistry schemes (e.g., bromine and iodine), which can influence the O3 simulation importantly.

4. Line 285, why not using the GEMM model in this work?

5. In the methodology section, more definition and explanation of YLL was needed. In the health analysis, what is the different meaning of analyzing attributable death and YLL, respectively?

6. Line 296, "Pop represents the exposed population in the age-gender-specific group in grid cell", but how to get these data for each grid wasn't mentioned in the context. E.g., did the age distribution of different provinces also come from yearbooks? Was the ratio of various age groups was the same for all the model grids?

7. In the model result evaluation, the authors used different statistical indicators for air pollutants and meteorological parameters because all used indicators were widely applied to both air pollutants and meteorological parameters in other studies. So the same indicators is suggested to be used for both, or the author needs to explain the reason.

8. Line 303, Table S4 does not have the information of LRI mortality rate

9. Line 441, based on the comparison between case3 or 4 and case2, it was concluded that the higher relative concentration change happened in July because of the faster response and high oxidative condition in this month. However, from the comparison of PM2.5 in case5 and case2, the larger concentration change also appears in January. For SO2 in case3 or 4, the decrease concentration in July is also not the largest. The decrease percentage is the largest, but it may due to the lower concentration in July.

The analysis is needed to be modified here.

10. The difference in Figure 3 and 4 were calculated by (Case2-Case3 or 4). Because the formula used in the previous analysis in Table 3 is (Case3 or 4 - Case2), so consistent formula was suggested to use in Figures 3 and 4.

11. Line 476, the author argued that the modest change of NO2 in central YRD (Shanghai, northern Zhejiang, and southern Jiangsu) caused an apparent enhancement of O3. But from Figure4 (Oct), the O3 in south Anhui also increased, but the change of NO2 here is much larger than that in Shanghai. How to definite the "modest"? More analysis and a better explanation are needed.

12. In the exposure analysis section (3.2.1), is there any basis for choosing these concentrations (35, 45, 55$\mu$g/m3) as interval value?

13. Line 610, "The fractions of both avoided deaths and YLL were clearly higher for Shanghai and part of Zhejiang, implying....." From which table or figure can you get this conclusion? Figure 9?

---

## Author Response (AR1)

**Main revisions and response to reviewers' comments**

**Journal:** Atmospheric Chemistry and Physics

**Manuscript No.:** acp-2020-818

**Title:** Air quality and health benefits from ultra-low emission control policy indicated by continuous emission monitoring: A case study in the Yangtze River Delta region, China

**Author:** Yan Zhang, Yu Zhao, Meng Gao, Xin Bo, Chris P. Nielsen

We thank very much for the valuable comments and suggestions from the reviewers, which help us improve our manuscript. The comments were carefully considered and revisions have been made in response to suggestions. Following are our point-by-point responses to the comments and corresponding revisions. **Please note that the line numbers mentioned following refer to the clean version of manuscript.**

**Reviewer #1[Report #1] (The Initial Submission stage):**

0. In this paper, authors evaluated the impact of including continuous emission monitoring (CEM) data in air quality simulation in comparison with emission estimated from traditional methods-"unit-level", using YRD as an example. Authors further evaluated the health benefits of applying "ultra-low emission control policy" on power plant and industry boilers in this region. The overall study design is clear, the methodology is appropriate, and the results are well-presented

**Response and revisions:**

We appreciate the reviewer's positive remarks.

1. As mentioned by authors, the monthly variations for sectors other than power plants were assumed to be the same. How does this assumption, to the authors' knowledge,

would affect the air quality simulation, especially the monthly disparity in R, NMB, NME?

**Response and revisions:**

We thank the reviewer's comment. We need to clarify that the monthly variations of sectors other than power sector were not the same as each other but the same as MEIC, the emission inventory at the national scale, as stated **in lines 235-236 in the revised manuscript**. We did not directly evaluate the bias from such assumption in this work. In our recent work (Zhao et al., 2019), a larger monthly variation in the emissions of black carbon aerosols was found for the central YRD region than that in MEIC, constrained by available ground observation. Limited improvement in modeling performance (indicated by R, NMB and NME) was consequently achieved. Therefore we believe the bias from temporal variation of emissions was insignificant. We have added the information **in lines 236-240 in the revised manuscript**.

2. How does the power plant emission estimated from CEM compared to the "level-based" method? And any monthly variations in the emission?

**Response and revisions:**

We thank the reviewer's comment. The detail comparison between emissions with and without CEMS data for power sector was provided in our previous work (Zhang et al., 2019). In this work, we summarize the difference for the YRD region **in Table 1 in the revised manuscript**. The emissions of $SO_2$, $NO_X$ and PM from power plants in Case 2 (with CEMS data ) were estimated as 427, 618 and 331 Gg smaller than those in Case 1 (without CEMS) for the YRD region.

The monthly variations of emissions based on CEMS data were also analyzed in our previous work (Zhang et al., 2019). In particular, larger monthly variation was found based on CEMS data compared to MEIC, as illustrated in Figure R1. We have added the information **in lines 258-259 in the revised manuscript**.

[Figure]

**Figure R1 The monthly variation of $NO_X$ (left) and $SO_2$ (right) emissions in this work based on CEMS data (Case 2) and MEIC. The data are taken from Zhang et al. (2019).**

3. How were the simulation results being compared with the observations? How many observation sites were included for comparison for the results presented in Table 2? Are there any patterns observed for different sites or function areas, e.g. roadside, rural, and urban?

**Response and revisions:**

We thank the reviewer's comment. The hourly concentrations were observed at 230 state-operated air quality monitoring stations within the YRD region, and the averages of hourly concentrations of those sites were compared with the simulations in Cases 1 and 2 in this work, as summarized in Table 2. We have added the information **in lines 365-368 in the revised manuscript**.

As most of the state-operated stations were located in urban regions, the patterns for different function areas could hardly be obtained through the observations at those sites. The information can be revealed when more observation data at rural or suburban regions (e.g., those from provincial-level stations) get available.

4. Line 443, "larger changes were found for $SO_2$ and $PM_{2.5}$ in summer." Author explained that possibly due to the "faster response of ambient concentrations to the changed emissions of air pollutants with shorter lifetimes in summer." But based on the validation results as shown in Table 2, the model performs relatively weaker in July (summer) than other months. Would this also contribute to the larger changes observed on line 443? If so, which one- the faster response or the poorer simulation contributes bigger to the changes?

**Response and revisions:**

We thank the reviewer's comment. As Cases 3-5 are hypothetical cases which evaluated the effect of changed emissions on the simulated air quality, we expected that the differences between those cases were less associated with the model performance for the base case (Case 2). Besides the lifetime issue, however, the biggest relative changes in $SO_2$ and $PM_{2.5}$ simulation for summer could also result from the lowest concentration in summer. To be more accurate, we have revised the text as "larger relative changes were found for $SO_2$ and $PM_{2.5}$ in summer" **in lines 481-482 in the revised manuscript**, and "the relatively low concentrations in summer also contributed to the largest percentage changes in $SO_2$ and $PM_{2.5}$ simulation for the season" **in lines 489-491 in the revised manuscript**. Please also see our response to Question 9 of Reviewer #3.

5. Line 539, "The numbers were larger than the estimate (6.9%) by Maji et al. (2018), which focused on 161 cities in China". What's the reason for the higher estimation in this study compared to the 161 cities, higher mortality rate or others?

**Response and revisions:**

We thank the reviewer's important comment. We expect the difference could result mainly from two issues. The first is the varied pollution levels for different study regions. This work evaluated the health effect from $PM_{2.5}$ exposure for the YRD region, while Maji et al. (2018) focused on 161 typical cities in China. As one of the

most developed and industrialized regions in China, YRD suffered higher $PM_{2.5}$ pollution level than the national average, leading to the larger fraction of premature death due to $PM_{2.5}$ exposure. The second is the choice of different baseline mortality rates. The disease-specific baseline incidence rates in Maji et al. (2018) were derived based on the national disease specific mortality in the dataset of GBD study for 2015, while the baseline age-gender-disease-specific mortality rates for the five diseases in this study were obtained from the Global Health Data Exchange database (GHDx). The numbers in the latter were commonly higher except for LRI, resulting in the higher estimates of death rates exposed to $PM_{2.5}$. We have added the information briefly **in lines 585-590 in the revised manuscript**.

**References:**

Zhang, Y., Bo, X., Zhao, Y., and Nielsen, C. P.: Benefits of current and future policies on emissions of China's coal-fired power sector indicated by continuous emission monitoring, Environ. Pollut., 251, 2019.

Zhao, X., Zhao, Y., Chen, D., Li, C., and Zhang, J.: Top-down estimate of black carbon emissions for city cluster using ground observations: A case study in southern Jiangsu, China, Atmos. Chem. Phys., 19, 2095-2113, 10.5194/acp-19-2095-2019, 2019.

**Reviewer #2 [Report #1] (the Interactive Discussion stage):**

0. Under the big pressure of air quality improvement, China has been conducting a series of measures to reduce the emissions of air pollutants, including the "ultra-low" emission policy for power and specific industrial sectors. Continuous emission monitoring system (CEMS) has been gradually installed and operated to examine the real emission status of individual plants. Besides, CEMS provided opportunities of improving the understanding of air pollutant emissions for atmospheric science community. Focusing on the YRD, one of the most economically developed region in

China, this paper presented an extended study based on the previously developed emission inventory of power sector using the CEMS measurement (Y. Zhang et al., 2019). The authors applied air quality model and evaluated the YRD emission inventory with CEMS incorporated. They further combined the air quality model and exposure-response model, and analyzed the benefit of ultra-low emission policy on air quality and health. It provides the evidence of emission data improvement and the environmental health implication of current air pollution control policy, thus it fits the scope of Atmos Chem Phys. In general the paper is well organized and clearly written. I would suggest its acceptance for publication, with minor revisions or discussions conducted on the following issues.

**Response and revisions:**

We appreciate the reviewer's positive remarks on our manuscript.

1. Abstract. Some sentences are unclear and should be rewritten. Lines 49-50: "11%, 7% and 2% of $SO_2$, $NO_X$ and PM" for which case?

**Response and revisions:**

We appreciate the reviewer's reminder. The language of abstract has been improved, and the numbers mentioned by the reviewer refer to Case 2, as we stated **in line 52 in the revised manuscript**.

2. Lines 155-157, did Gao et al. (2018) (and some other studies maybe) include the CEMS data for emission inventory development and stress the ultra-low emission policy?

**Response and revisions:**

We thank the reviewer's comment. Gao et al. (2018) (and most other studies at the national scale) applied MIX (Li et al., 2017) or MEIC (Zheng et al., 2018), and CEMS

data were not comprehensively incorporated. The target year in Gao et al. (2018) is 2013, in which the ultra-low emission policy was not conducted yet.

3. Lines 226-234. It seems that you apply the inventory by Xia et al. (2016) but the spatiotemporal distribution of MEIC? Why not use MEIC directly?

**Response and revisions:**

We thank the reviewer's comment. We applied this method basically for two reasons. First, as one of our previous studies, Xia et al. (2016) calculated the annual emissions by province and species for China, using a similar "bottom-up" method with MEIC. Given the difference choice of emission factors for certain sources, there were differences in the amount of emissions for some provinces and sectors between the two inventories. Second, MEIC provided the emission data of total industry but did not report the specific information for industrial boilers, cement or iron & steel factories when this work was conducted. As the ultra-low emission policy was assumed to be conducted for industrial boilers, cement and iron & steel factories in this work (Case 4), we needed to calculate the emissions exactly for the same categories in the base case (Case 2). In this work, therefore, we applied the provincial-level emission data by Xia et al. (2016) and obtained the gridded data according to the spatial distribution of emissions by MEIC.

4. Lines 251-253. Some information is missing here. Case 4 assumes both power and industrial boilers would meet the ultra-low emission limit. Do the two sectors share the same limit? In your previous work (Y. Zhang et al., 2019) you analyzed the emission limit for power sector with CEMS incorporated, but how about references for industrial boilers? Description should be given here.

**Response and revisions:**

We appreciate the reviewer's important remarks and acknowledge the information was unclear in the original submission. In Case 4 we assumed industrial boilers, cement, and iron & steel factories would meet the requirement of ultra-low emission policy. The limits of flue gas concentrations were determined according to available national or local ultra-low emission standards issued recently (Yang et al., 2021). Thus the limits vary by sector and are different from those for power sector. We have summarized the ultra-low emission limits and standards by sector **in a new Table S2 in the revised supplement**, and added the information in **in lines 267-271 in the revised manuscript**.

5. Table 1 seems unnecessary, while the emission data in difference cases are more important (Table S2). Could you combine Table 1 and Table S2?

**Response and revisions:**

We thank the reviewer's comment. We combined Table 1 and Table S2 in the original submission as **a new Table 1 in the revised manuscript**.

6. Line 326-328. The difference for $SO_2$ was 10%, did this contradict the statement in lines 45-46 in the abstract: ": $SO_2$, $NO_2$, $O_3$ and $PM_{2.5}$ concentrations compared to those of Case 2, our base case for policy comparisons, were less than 7% for all pollutants"?

**Response and revisions:**

We thank the reviewer's comment. The 10% difference for $SO_2$ was between Case 1 and Case 2, while the statement in abstract indicated the difference between Case 2 and Case 3. Thus they did not contradict each other.

7. Lines 359-360. I cannot quite understand the numbers. You are comparing NMBs and NMEs for two cases, then what did the two groups of numbers exactly stand for?

**Response and revisions:**

We thank the reviewer's comment. To avoid the confusion, the sentence was rewritten as: "The NMBs between the simulation and observation for the two cases ranged from -34.5% to -6.4%, and NMEs from 23.1% to 37.1%, respectively" **in lines 389-390 in the revised manuscript**. The wrong numbers has also been corrected in the sentence.

8. Line 367. Is it commonly known that the YRD is under VOC-limited regime in terms of O3 formation? At least some literatures should be provided here.

**Response and revisions:**

We thank the reviewer's comment. Some recent studies revealed or confirmed that most of YRD was under VOC-limited for $O_3$ formation (Wang et al., 2019; Yang et al., 2021). We revised the sentences as: "As most of YRD was identified as a VOC-limited region for $O_3$ formation (Wang et al., 2019; Yang et al., 2021)" **in lines 397-398 in the revised manuscript**.

9. Lines 435-437. Did the "ultra-low" emission policy include the limit of VOCs? Or did CEMS include the information of VOCs? Some explanations should be given here.

**Response and revisions:**

We thank the reviewer's comment. CEMS does not report VOC concentration in the flue gas, and current "ultra-low emission" policy does not include VOC limits, either. We have added the information **in lines 474-475 in the revised manuscript**.

10. Lines 539-540: Any reason could be provided?

**Response and revisions:**

We appreciate the reviewer's comment and it is the same as the Question 5 from Reviewer #1. We expect the difference could result mainly from two issues. The first is the varied pollution levels for different study regions. This work evaluated the health effect from $PM_{2.5}$ exposure for the YRD region, while Maji et al. (2018) focused on 161 typical cities in China. As one of the most developed and industrialized regions in China, the YRD suffered higher $PM_{2.5}$ pollution level than the national average, leading to the larger fraction of premature death due to $PM_{2.5}$ exposure. The second is the choice of different mortality rates. The disease-specific baseline incidence rates in Maji et al. (2018) were derived based on the national disease specific mortality in the dataset of GBD study for 2015, while the baseline age-gender-disease-specific mortality rates for the five diseases in this study were obtained from the Global Health Data Exchange database (GHDx). The numbers in the latter were commonly higher except for LRI, resulting in the higher estimates of death rates exposed to $PM_{2.5}$. We have added the information briefly **in lines 585-590 in the revised manuscript**.

11. Language issues. Line 119: "environmental concentrations"? Line 125: delete the word "itself"; Line 504: change "sectors" to "boilers"; Lines 531-534: rewrite the sentence

**Response and revisions:**

We thank the reviewer's reminder and correct the language errors.

**Reference**

Gao, M., Beig, G., Song, S., Zhang, H., Hu, J., Ying, Q., Liang, F., Liu, Y., Wang, H., Lu, X., Zhu, T., Carmichael, G. R., Nielsen, C. P., and McElroy, M. B.: The impact of

power generation emissions on ambient PM$_{2.5}$ pollution and human health in China and India, Environ. Int., 121, 250-259, 10.1016/j.envint.2018.09.015, 2018.

Li, M., Zhang, Q., Kurokawa, J.I., Woo, J.H., He, K., Lu, Z., et al. MIX: a mosaic Asian anthropogenic emission inventory under the international collaboration framework of the MICS-Asia and HTAP, Atmos. Chem. Phys., 17, 935-963, 10.5194/acp-17-935-2017, 2017.

Wang, N., Lyu, X., Deng, X., Huang, X., Jiang, F., and Ding, A.: Aggravating O$_3$ pollution due to NOx emission control in eastern China, Sci. Total Environ., 677, 732-744, 2019.

Yang, J., Zhao, Y., Cao, J., and Nielsen, C.: Co-benefits of carbon and pollution control policies on air quality and health till 2030 in China, Environ. Int., 152, 106482, 10.1016/j.envint.2021.106482, 2021.

Yang, Y., Zhao, Y., Zhang, L., Zhang, J., Huang, X., Zhao X., et al.: Improvement of the satellite-derived NOx emissions on air quality modeling and its effect on ozone and secondary inorganic aerosol formation in the Yangtze River Delta, China. Atmos. Chem. Phys., 21, 1191-1209, 10.5194/acp-21-1191-2021, 2021.

Zheng, B., Tong, D., Li, M., Liu, F., Hong, C., Geng, G., et al. Trends in China's anthropogenic emissions since 2010 as the consequence of clean air actions. Atmos. Chem. Phys., 18, 14095-14111, 10.5194/acp-18-14095-2018, 2018.

**Reviewer #3 [Report #2] (the Interactive Discussion stage):**

0. This paper evaluated the potential benefit of the ultra-low emission policy on both air quality and human health in the YRD region. No novel technique was developed, or new scientific finding was reported. The results can still provide some scientific reference for related emission control policy and health burden caused by air pollution over the YRD region. Overall, this paper is well written, but more description in the

methodology is still needed. A major revision is suggested, and my specific comments are listed as follows.

**Response and revisions:**

We appreciate the reviewer's remarks and have revised the manuscript according to the reviewer's specific comments, as summarized below.

1. Line 55, in the abstract section, "874 years", but according to on paper Table6, it should be "8744 years of life loss".

**Response and revisions:**

We thank the reviewer's reminder and the error has been corrected.

2. In the methodology section, please generally introduce the method of how to incorporate the CEMS data and cite the references which have the detailed description.

**Response and revisions:**

We appreciate the reviewer's important comment. The main principle of the method incorporating the CEMS data has been described and the reference has been provided **in lines 251-260 in the revised manuscript**: "Besides the commonly used method, Y. Zhang et al. (2019) developed a new method of examining, screening and applying CEMS data to improve the estimates of power sector emissions. CEMS data were collected for over 1000 power units, including operation condition, monitoring time, flue gas flow, and hourly concentrations of $SO_2$, NOx and PM. The emissions of individual unit were calculated based on the hourly concentrations of air pollutants obtained from CEMS and the theoretical flue gas volume estimated based on the unit-level information mentioned above. Compared to MEIC, a larger monthly

variation in emissions was found based on the online emission monitoring. Details can be found in Y. Zhang et al. (2019)."

3. Why still using the old version of the CMAQ model? The current CMAQ model (v5.2 or v5.3) has incorporated several trace gas chemistry schemes (e.g., bromine and iodine), which can influence the O3 simulation importantly.

**Response and revisions:**

We thank the reviewer's important comment. We acknowledge that application of the old version of CMAQ is a limitation in this work. In our recent work (Lu et al., 2020), we tested the model performances for the YRD region with different versions of CMAQ, and found the impact of CMAQ version on simulation for difference species was inconclusive. Generally, the $PM_{2.5}$ simulation was improved with newer version, but the $O_3$ simulation was not, particularly for the periods with relatively low concentrations. The test revealed the necessity of further intercomparison and evaluation studies for the region. We have added the discussions **in lines 419-425 in the revised manuscript**.

4. Line 285, why not using the GEMM model in this work?

**Response and revisions:**

We appreciate the reviewer's important comment. Indeed the choice of health model (or Concentration-Response function, C-R function) is of great impact on the result of health effect analysis. In our most recent work, actually, we compared the premature mortalities estimated with IER and GEMM in 2030 energy saving and emission control scenarios for China (Yang et al., 2021). The larger GEMM hazard ratio as well as higher baseline mortality rates resulted in higher $PM_{2.5}$-related mortalities than IER. Therefore, application of IER got a relatively conservative estimate for the health effect of air pollution and the benefit of emission controls. As the range of $PM_{2.5}$

exposure in China could be larger than that considered in GEMM (84 μg/m$^3$), we believed IER would applicable for the country. It has been relatively well developed for mortality estimation and has been widely used in quantifying the impact of environmental policies and air quality standards on health burden (Li et al. 2019; Yue et al. 2020; Zheng et al. 2019). We have added the explanation **in lines 292-297 in the revised manuscript**.

5. In the methodology section, more definition and explanation of YLL was needed. In the health analysis, what is the different meaning of analyzing attributable death and YLL, respectively?

**Response and revisions:**

We thank the reviewer's comment. YLL represents the years of life lost because of premature death from a particular cause or disease. It is calculated from the number of deaths multiplied by a standard life expectancy at the age at which death occurs. Death rates could not provide a comprehensive picture of the burden that deaths impose on the population, thus YLL caused by PM$_{2.5}$ exposure was estimated in this study to help describe the extent to which the lives of people exposed to air pollution were cut short. We have added such information **in lines 299-303 in the revised manuscript**.

6. Line 296, "Pop represents the exposed population in the age-gender-specific group in grid cell", but how to get these data for each grid wasn't mentioned in the context. E.g., did the age distribution of different provinces also come from yearbooks? Was the ratio of various age groups was the same for all the model grids?

**Response and revisions:**

We thank the reviewer's reminder. The gender distributions of different provinces were obtained from provincial yearbooks. As the high-resolution spatial pattern of age

structure was unavailable when the study was conducted, we assumed the age structure was the same for all the model grids (Gao et al., 2018). We have added the information **in lines 326-329 in the revised manuscript**.

7. In the model result evaluation, the authors used different statistical indicators for air pollutants and meteorological parameters because all used indicators were widely applied to both air pollutants and meteorological parameters in other studies. So the same indicators are suggested to be used for both, or the author needs to explain the reason.

**Response and revisions:**

We thank the reviewer's comment. For meteorological parameters we followed Emery et al. (2001) and applied the main statistical indicators and benchmarks suggested in the study. For air quality modeling, R, NMB, and NME are mostly applied for comparison between simulation and observation, thus they were adopted in this work.

8. Line 303, Table S4 does not have the information of LRI mortality rate

**Response and revisions:**

We thank the reviewer's reminder. LRI is a common disease among young children, thus we applied uniform mortality rates regardless of age. The LRI baseline mortalities were 13.7 and 11.4 cases per 100,000 for male and female, respectively. We have added the information **in the caption of Table S4**.

9. Line 441, based on the comparison between Case 3 or 4 and Case 2, it was concluded that the higher relative concentration change happened in July because of the faster response and high oxidative condition in this month. However, from the comparison of $PM_{2.5}$ in Case 5 and Case 2, the larger concentration change also

appears in January. For $SO_2$ in Case 3 or 4, the decrease concentration in July is also not the largest. The decrease percentage is the largest, but it may due to the lower concentration in July. The analysis is needed to be modified here.

**Response and revisions:**

We thank and agree the reviewer's comment. The changes in the absolute concentrations of $SO_2$ and $PM_{2.5}$ were not always the largest for summer, but those in percentages were. To be more accurate, we have revised the text as "larger relative changes were found for $SO_2$ and $PM_{2.5}$ in summer" **in lines 481-482 in the revised manuscript**, and "the relatively low concentrations in summer also contributed to the largest percentage changes in $SO_2$ and $PM_{2.5}$ simulation for the season" **in lines 489-491 in the revised manuscript**.

10. The difference in Figure 3 and 4 were calculated by (Case 2-Case 3 or 4). Because the formula used in the previous analysis in Table 3 is (Case 3 or 4-Case 2), so consistent formula was suggested to use in Figures 3 and 4.

**Response and revisions:**

We thank the reviewer's reminder. We have revised Figures 3 and 4 using the consistent formulas with Table 3. The captions of the two figures have also been corrected accordingly.

11. Line 476, the author argued that the modest change of $NO_2$ in central YRD (Shanghai, northern Zhejiang, and southern Jiangsu) caused an apparent enhancement of $O_3$. But from Figure 4 (Oct), the $O_3$ in south Anhui also increased, but the change of $NO_2$ here is much larger than that in Shanghai. How to definite the "modest"? More analysis and a better explanation are needed.

**Response and revisions:**

We appreciate the reviewer's important comment. We acknowledge that the original explanation did not cover all the important information, and the word "modest" was confusing. As mentioned earlier in the paper, most of YRD was identified as a VOC-limited region for $O_3$ formation. Here the modeling results show that the central YRD (Shanghai, northern Zhejiang, and southern Jiangsu) was the most influenced by the mechanism. Compared to other areas (e.g., southern Anhui as pointed by the reviewer), the relatively less reduction in $NO_X$ (and thereby $NO_2$) would lead to significant enhancement of $O_3$ (note much more reduction in $NO_2$ resulted in similar enhancement of $O_3$ in southern Anhui for October). The comparison implies that the $O_3$ formation in central YRD was more sensitive to $NO_X$ emission abatement than other VOC-limited regions in YRD. Therefore, more efforts on VOC emission abatement would be required for $O_3$ pollution control in central YRD. We have revised the explanation, as shown **in lines 516-523 in the revised manuscript**.

12. In the exposure analysis section (3.2.1), is there any basis for choosing these concentrations (35, 45, 55 µg/m$^3$) as interval value?

**Response and revisions:**

We thank the reviewer's comment. The main reason is that 35 µg/m$^3$ is the annual PM$_{2.5}$ concentration limit in the current National Ambient Air Quality Standard (NAAQS) for China. We thus apply an interval of 10µg/m$^3$ based on that limit, as the C-R function is usually expressed as %/(10 µg/m$^3$). We have added the information **in lines 545-547 in the revised manuscript**.

13. Line 610, "The fractions of both avoided deaths and YLL were clearly higher for Shanghai and part of Zhejiang, implying..." From which table or figure can you get this conclusion? Figure 9?

**Response and revisions:**

We thank the review's comment. The information can be obtained from Tables 5 and 6. As can be inferred from the two tables, the fractions of Shanghai and Zhejiang to total YRD for both avoided deaths and YLL increased clearly from Case 3 to Case 4. We have added the information **in lines 661-664 in the revised manuscript** and deleted the phrase "part of" to avoid confusion.

**Reference**

Emery, C., Tai, E., and Yarwood, G.: Enhanced meteorological modeling and performance evaluation for two Texas episodes, Report to the Texas Natural Resources Conservation Commission, prepared by ENVIRON, International Corp, Novato, CA, 2001.

Gao, M., Beig, G., Song, S., Zhang, H., Hu, J., Ying, Q., Liang, F., Liu, Y., Wang, H., Lu, X., Zhu, T., Carmichael, G. R., Nielsen, C. P., and McElroy, M. B.: The impact of power generation emissions on ambient $PM_{2.5}$ pollution and human health in China and India, Environ. Int., 121, 250-259, 10.1016/j.envint.2018.09.015, 2018.

Li, M., Zhang, D., Li, C.-T., Selin, N.E., and Karplus, V.J.: Co-benefits of China's climate policy for air quality and human health in China and transboundary regions in 2030, Environ. Res. Lett., 14, 10.1038/s41558-018-0139-4, 2019.

Lu, Y., Zhao, X., and Zhao, Y.: The comparison and evaluation of air pollutant simulation for the Yangtze River Delta region with different versions of air quality model, Environ. Monit. Forewarn., 12, 10.3969/j.issn.1674- 6732.2020.03.001, 2020 (in Chinese).

Yang, J., Zhao, Y., Cao, J., and Nielsen, C.: Co-benefits of carbon and pollution control policies on air quality and health till 2030 in China, Environ. Int., 2021 (in press).

Yue, H., He, C., Huang, Q., Yin, D., and Bryan, B. A.: Stronger policy required to substantially reduce deaths from $PM_{2.5}$ pollution in China, Nat. Commun., 11, 1462, 10.1038/s41467-020-15319-4, 2020.

Zhang, Y., Bo, X., Zhao, Y., and Nielsen, C. P.: Benefits of current and future policies on emissions of China's coal-fired power sector indicated by continuous emission monitoring, Environ. Pollut., 251, 415-424, 2019.

Zheng, H., Zhao, B., Wang, S., Wang, T., Ding, D., Chang, X., Liu, K., and Xing, J.: Transition in source contributions of $PM_{2.5}$ exposure and associated premature mortality in China during 2005-2015, Environ. Int. 132, 105111, 10.1016/j.envint.2019.105111, 2019.